# Scaling Collapse Reveals Universal Dynamics in Compute-Optimally Trained Neural Networks

**Shikai Qiu** [1][†] **Lechao Xiao** [2] **Andrew Gordon Wilson** [1] **Jeffrey Pennington** [2] **Atish Agarwala** [2]

## Abstract

What scaling limits govern neural network training dynamics when model size and training time grow in tandem? We show that despite the complex interactions between architecture, training algorithms, and data, compute-optimally trained models exhibit a remarkably precise universality. Specifically, loss curves from models of varying sizes collapse onto a single universal curve when training compute and loss are normalized to unity at the end of training. With learning rate decay, the collapse becomes so tight that differences in the normalized curves across models fall below the noise floor of individual loss curves across random seeds, a phenomenon we term *supercollapse*. We observe supercollapse across learning rate schedules, datasets, and architectures, including transformers trained on next-token prediction, and find it breaks down when hyperparameters are scaled suboptimally, providing a precise and practical indicator of good scaling. We explain these phenomena by connecting collapse to the power-law structure in typical neural scaling laws, and analyzing a simple yet surprisingly effective model of SGD noise dynamics that accurately predicts loss curves across various learning rate schedules and quantitatively explains the origin of supercollapse.

## 1. Introduction

As machine learning systems grow in scale, accurate predictive models of their training dynamics become increasingly valuable, both for interpreting costly experiments and for designing robust, efficient training pipelines (Wortsman et al.,

[†]Work done partly during an internship at Google DeepMind. [1]New York University [2]Google DeepMind. Correspondence to: Shikai Qiu <sq2129@nyu.edu>, Atish Agarwala <thetish@google.com>.

*Proceedings of the 42^{nd} International Conference on Machine Learning*, Vancouver, Canada. PMLR 267, 2025. Copyright 2025 by the author(s).

2023; Achiam et al., 2023; Xiao, 2024). While the complexity of modern architectures, optimizers, and datasets often renders exact, first-principles analyses intractable for any individual model, recent work shows that some key aspects of training are predictable when we focus on their scaling behavior across a family of models. Examples include empirical power-law relations linking optimal final loss, model size, dataset size, and compute budget under compute-optimal training, known as neural scaling laws (Hestness et al., 2017; Kaplan et al., 2020; Sharma & Kaplan, 2022; Hoffmann et al., 2022), as well as hyperparameter transfer from small to large models based on infinite-width or depth limits of training dynamics (Yang et al., 2021; Bordelon et al., 2023; Everett et al., 2024; Bordelon et al., 2024c).

In this work, we show the entire training process follows highly predictable scaling, beyond final losses and optimal hyperparameters. We find that the entire loss curves of compute-optimally trained models exhibit a precise scaling symmetry, collapsing onto a single universal curve across models after a simple normalization. Learning rate decay amplifies this effect dramatically, producing what we call *supercollapse*: collapse so tight that cross-scale differences fall below the noise floor of individual loss curves due to random seeds. Figure 1 (a-d) summarizes these results.

These findings advance our understanding in two key ways. First, while Kaplan et al. (2020, Figure 11) found the loss curves roughly follow a sum of power laws, we identify that loss curves follow a universal shape with far greater precision. For typical learning rate schedules, this shape deviates from simple power laws and may not admit any obvious functional form. Second, our work provides compelling empirical evidence for a well-defined joint scaling limit where model size and training time grow together under compute-optimal allocation. This limit contrasts with traditional infinite-width or depth limits that fix training duration (Yang & Hu, 2021; Bordelon & Pehlevan, 2022). While these theories predict initial dynamical consistency, accumulating finite-size effects lead to gradual divergence as training progresses, as demonstrated by Vyas et al. (2023). In contrast, the collapse we observe reveals a joint scaling limit that preserves consistency throughout training, precisely the regime relevant for practical large-scale training.

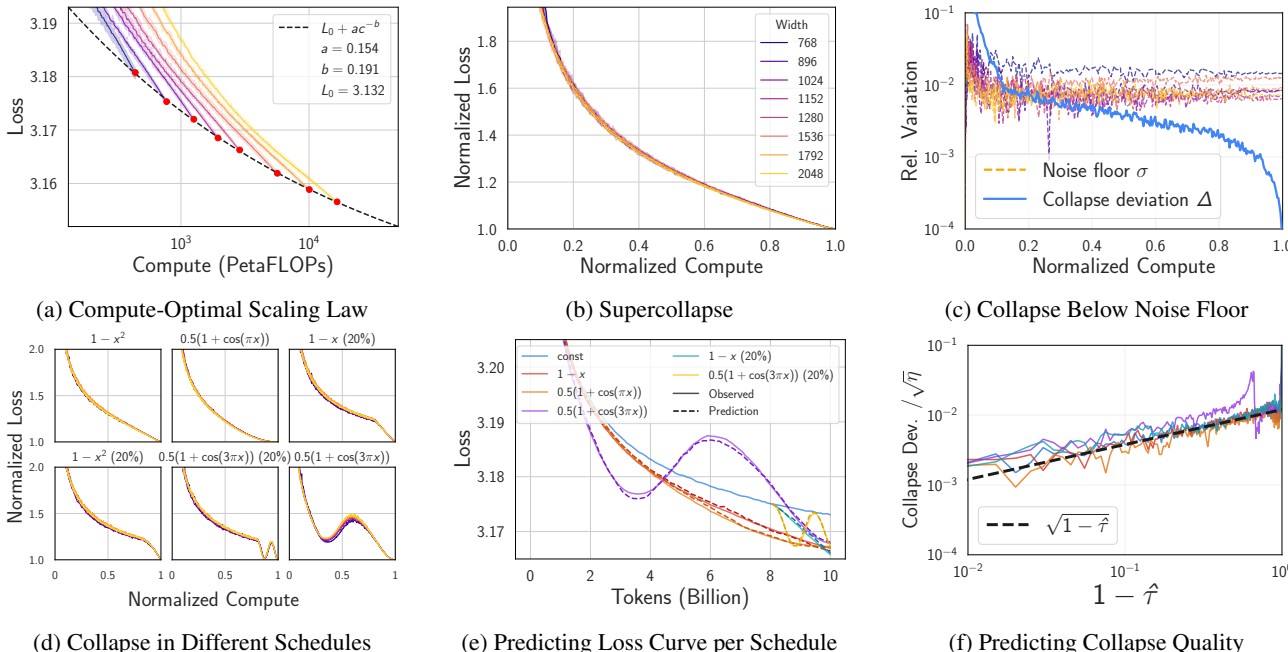

(a) Compute-Optimal Scaling Law    (b) Supercollapse    (c) Collapse Below Noise Floor

(d) Collapse in Different Schedules    (e) Predicting Loss Curve per Schedule    (f) Predicting Collapse Quality

Figure 1: **Scaling collapse of compute-optimal transformer loss curves and its explanation through a model of SGD noise dynamics.** **(a)** Compute-optimal loss curves and fitted scaling law on CIFAR-5M, using a linear learning rate decay schedule. **(b)** Normalized reducible loss curves collapse onto a single universal curve independent of model size, with both final compute and reducible loss normalized to unity. **(c)** Collapse deviation $\Delta$ (cross-model variation of normalized loss) falls below per-model noise floor $\sigma$ (variation of reducible loss across random seeds) for much of training, a phenomenon we term *supercollapse*. **(d)** Supercollapse occurs during the decay phase of various learning rate schedules, each producing its own universal curve. To explain these phenomena, we show that a simple model of SGD noise dynamics **(e)** accurately predicts loss curves for different schedules across model scales (Section 3.2) and **(f)** quantifies how learning rate decay improves the collapse due to the predicted scaling $\Delta \propto \sqrt{\eta(1 - \hat{\tau})}$, where $\eta$ is the instantaneous learning rate and $\hat{\tau}$ is normalized gradient flow time (Section 3.3). We observe supercollapse in other arhitectures and datasets (Figure 4).

We provide an elementary theoretical analysis that reveals the key mechanisms behind this precise collapse. We first show that for loss curves following typical neural scaling laws, collapse occurs precisely when models are trained for constant multiples of their compute-optimal horizons (Section 3.1). We then analyze a simple theoretical model of the SGD noise dynamics that predicts loss curves under a variety of learning rate schedules remarkably well (Section 3.2), and explains two key observations: why normalized curves retain universal form despite losing their power-law structure, and how learning rate decay suppresses variance to produce supercollapse (Section 3.3).

Beyond theoretical interest, supercollapse provides a practical scaling diagnostic, as we find that deviations from collapse can signal misconfigured scaling choices, such as suboptimal scaling of learning rate and data (Figure 4). Overall, our results suggest supercollapse provides a novel, powerful tool to study scaling. Our code can be found here.

## 2. Empirical Observations

We demonstrate our main empirical findings in this section, independently on multiple tasks and architectures which can be studied even in academic settings.

### 2.1. Experiment Setup

In each task, we train a sequence of models with increasing compute, scaling hyperparameters such as data, initialization, and learning rate with the model. We refer to a sequence of training configurations as a scaling ladder. We provide further experimental details in Appendix A. We focus on width scaling, where hyperparameter transfer is most well-studied, but find scaling transformer depth leads to similar results in Appendix B, suggesting our observations may generalize to more general scaling ladders where width, depth, batch size, weight decay, etc. can be co-scaled.

**Transformers Next-Token Prediction.** We consider two next-token prediction tasks: 1) CIFAR-5M (Nakkiran et al., 2020), a dataset of 6M generated CIFAR-like images, and 2) Lichess, a collection of chess games recorded in algebraic chess notation where the goal is to predict the next move in the game. Our scaling ladder includes models with about 10M to 80M parameters, approximately log-uniformly spaced, by scaling the width (embedding dimensions) from 768 to 2048 and fixing the number of blocks to 3. All models use $\mu$P (Yang & Hu, 2021; Yang et al., 2021) for initialization and learning rates, and are trained with Adam.

**MLPs on Power-Law Fourier Features.** To investigate

other architectures and training objectives, we train 7-layer MLPs with varying widths from 384 to 2048 on a synthetic regression task. The target function has a power-law Fourier spectrum, designed to elicit the power-law scaling laws observed in natural data. We count each example as 1 token.

## 2.2. Estimating Compute-Optimal Scaling Laws

Let $L(t, p, \omega)$ be the test loss after $t$ tokens (proportional to steps) for a model with $p$ parameters trained with random seed $\omega$. We estimate the compute-optimal training horizon in tokens for a $p$-parameter model as $t^\star(p) = (p/p_0)^\gamma$, where $\gamma$ is the data exponent, by extracting the Pareto frontier of expected loss (estimated using 5 seeds) vs. compute under a constant learning rate schedule, following a procedure similar to Approach 1 in Hoffmann et al. (2022), with compute estimated as $c = 6tp$ FLOPs (Kaplan et al., 2020). We reuse the same $t^\star(p)$ as the training horizon for other learning rate schedules, which prior work suggests is optimal up to a constant factor (Pearce & Song, 2024). For each task and schedule, we fit the resulting compute-optimal scaling law using the form $L_0 + ac^{-b}$ (Figure 1a), for constants $L_0, a, b \geq 0$. Following Sharma & Kaplan (2022) and Hoffmann et al. (2022), we refer to $L_0$ as the estimated irreducible loss. Using the best-fit $L_0$, we define the reducible loss curve $\mathcal{L}(t, p, \omega) = L(t, p, \omega) - L_0$. We detail the procedure for fitting the compute-optimal training horizon in Appendix C. "Compute-optimal" here primarily refers to the choice of training horizon, not of all hyperparameters.

## 2.3. Scaling Collapse of Compute-Optimal Loss Curves

The loss curves for different model sizes cover varying ranges of compute and loss values, but appear to follow a consistent shape, which motivates us to affinely rescale them to the *normalized loss curve* $\ell$ given by

$$\ell(x, p, \omega) = \frac{L(xt^\star(p), p, \omega) - \hat{L}}{L(t^\star(p), p, \omega) - \hat{L}}, \quad x \in [0, 1], \quad (1)$$

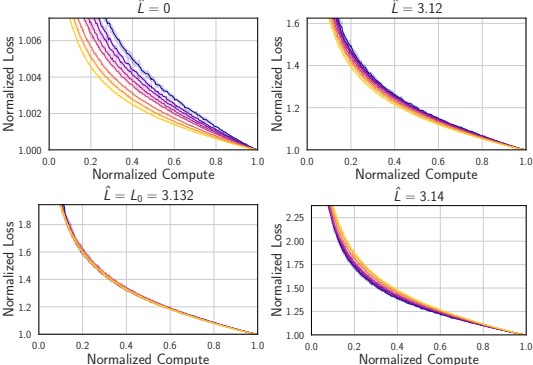

Figure 2: **Subtracting irreducible loss leads to the best collapse.** Setting $\hat{L}$ to values far from $L_0$ breaks the collapse on CIFAR-5M.

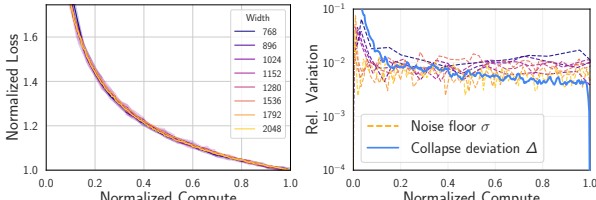

Figure 3: **Collapse with a constant LR schedule.** (Left) Estimated mean and 90% confidence interval (shaded) of the normalized loss curves. (Right) $\Delta$ is comparable to $\sigma$ without LR decay.

for some offset $\hat{L}$. We refer to $x$ as the normalized compute. Note the denominator uses the stochastic final loss value specific to the random seed. We set $\hat{L} = L_0$ to subtract the estimated irreducible loss that bottlenecks the asymptotic performance, leading to $\ell(x, p, \omega) = \frac{\mathcal{L}(xt^\star(p), p, \omega)}{\mathcal{L}(t^\star(p), p, \omega)}$.

Remarkably, we observe that the family of normalized loss curves is nearly identical across $p$, revealing equal rates of relative progress (Figure 1b). We say these curves *collapse*, as the phenomenon resembles the ubiquitous scaling collapse found in statistical physics, theoretical biology, and other sciences, where observables from systems of different sizes collapse onto a single curve after appropriate rescaling (see Appendix D for further discussion). We found setting $\hat{L} = L_0$ achieves the best collapse (Figure 2).

## 2.4. Quantifying the Collapse Quality

We quantify the quality of collapse using the *collapse deviation* $\Delta$, defined as:

$$\Delta(x) = \frac{\mathbb{V}_{p,\omega}[\ell(x, p, \omega)]^{1/2}}{\mathbb{E}_{p,\omega}[\ell(x, p, \omega)]}, \quad (2)$$

where $\mathbb{E}_{p,\omega}$ and $\mathbb{V}_{p,\omega}$ denote the expectation and variance over the random seed and the empirical distribution of model size $p$ in the scaling ladder (approximately log-uniformly distributed). The collapse deviation measures the relative variation of the normalized curves across $p$. For perspective, we compare it to the per-model (relative) noise floor:

$$\sigma(x, p) = \frac{\mathbb{V}_\omega[\mathcal{L}(xt^\star(p), p, \omega)]^{1/2}}{\mathbb{E}_\omega[\mathcal{L}(xt^\star(p), p, \omega)]}, \quad (3)$$

which measures the relative fluctuation in the reducible loss curve for each model size $p$ across random seeds.

By the definition of $\ell$, $\Delta(1) = 0$ always. As seen in Figure 3, for a constant learning rate, $\Delta(x)$ quickly rises to a level comparable to $\sigma(x, p)$, and remains at that level for most $x < 1$. This observation shows that variations in the normalized curves arise primarily from seed-to-seed fluctuations rather than model-to-model differences, quantitatively demonstrating that the observed collapse is non-trivial.

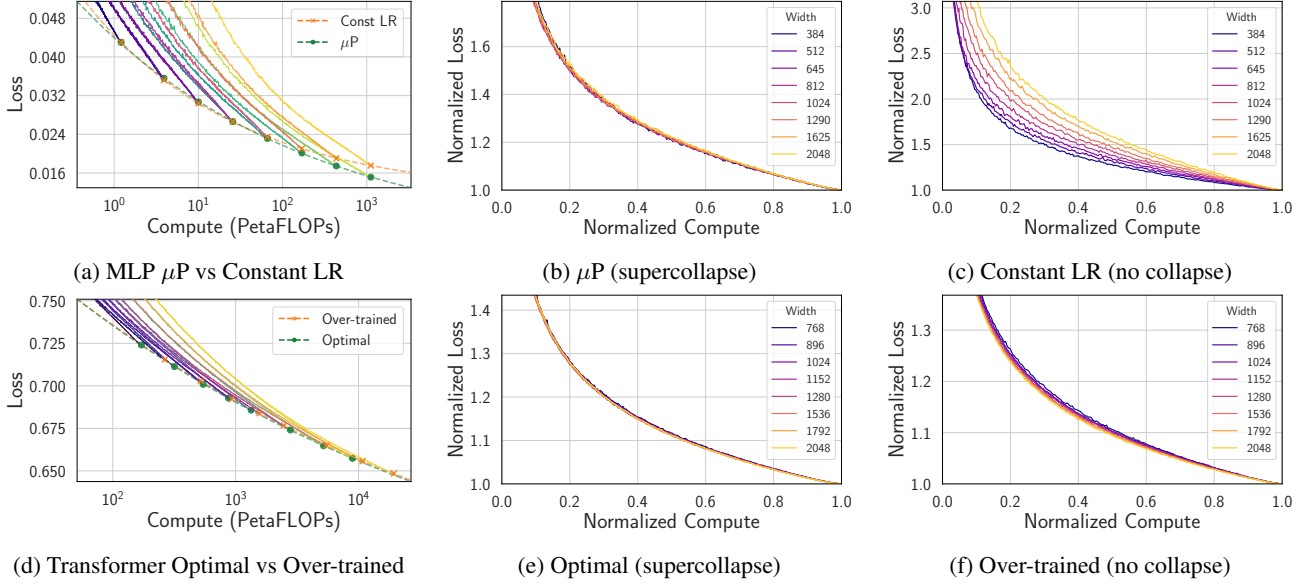

Figure 4: **Collapse provides a practical indicator of good scaling, as suboptimally scaling key hyperparameters breaks the collapse.** With the default setup, we observe supercollapse in MLP regression (**b**) and transformer trained on chess (**e**), but even changes that only lead to minor worsening in the scaling law can manifest as significant disruption to the collapse. (**Top**) Replacing $\mu$P with a constant learning rate cross models for MLPs. (**Bottom**) Increasing the data exponent $\gamma$ from estimated compute-optimal value 1.02 to 1.2 for Transformers trained on chess. We perform a separate power-law fit to determine the value $L_0$ for each scaling ladder.

## 2.5. Supercollapse: Consistency Below the Noise Floor

Remarkably, with learning rate decay, we find that the collapse deviation is less than the noise floor for a significant fraction of training; that is, $\Delta(x) < \sigma(x, p)$ for $x > 1 - \delta$ for some moderate $\delta$ as large as 0.5 (Figure 1c). We refer to this stronger form of collapse as *supercollapse* (in contrast to the collapse in Figure 3). Supercollapse appears in the decay phase of all tested learning rate schedules that decay to zero (Figure 1d). All schedules are defined in terms of relative training fraction, i.e., the learning rate is a fixed function of the normalized compute $x$ across model sizes.

Under supercollapse, self-normalized loss curves from different models collapse better than our ability to predict any individual model's loss. Normalizing by the final loss of the particular realization of the stochastic loss curve is key to supercollapse, which reduces variance by exploiting correlations at different times along a single optimization trajectory. We explain this mechanism in detail in Section 3.3.

## 2.6. Suboptimal Scaling Breaks Supercollapse

Supercollapse provides a practical method for comparing inherently noisy training loss curves across model scales with precision that exceeds naive noise floor estimates, without the need for expensive multi-seed experiments typically required to obtain equally clean signals. This comparison can provide valuable diagnostic information about scaling where the ability to distinguish small signal from noise is often crucial (Xiao, 2024), which we now demonstrate.

**Model Parameterization.** Carefully parameterizing the model, i.e., scaling the initialization, learning rate, and possibly other hyperparameters as model size increases, is crucial for achieving stable and efficient training at scale (Yang et al., 2021; Bordelon et al., 2023; 2024c; Everett et al., 2024). When models are trained in the wrong parameterization, we expect the loss curves not to collapse due to a lack of consistent training dynamics across scales. Using the MLP setup, we show that replacing $\mu$P with a constant learning rate across widths breaks the collapse (Figure 4, top row). Remarkably, the normalized loss curves expose inconsistent dynamics even at small scales where the final losses are virtually identical between constant and $\mu$P scaling, demonstrating that the collapse is a more sensitive probe of scaling behavior than final performance alone.

**Compute-Optimal Data Exponent.** For language models, Kaplan et al. (2020) showed that compute-optimal training corresponds to training each model to a fixed multiple of its converged loss. If this principle generalizes to our setting, the data exponent $\gamma$ should match the compute-optimal value for collapse to occur. For example, when $\gamma$ exceeds the optimal value, larger models will make more rapid relative initial progress but decelerate later as a function of normalized compute, causing their normalized curves to shift downward. We indeed find this shift in Figure 4 (bottom row). This sensitivity suggests a novel application: rather than fitting power laws to sparse points on the Pareto frontier, one could tune $\gamma$ to maximize collapse quality, leveraging the full statistical power of entire loss curves.

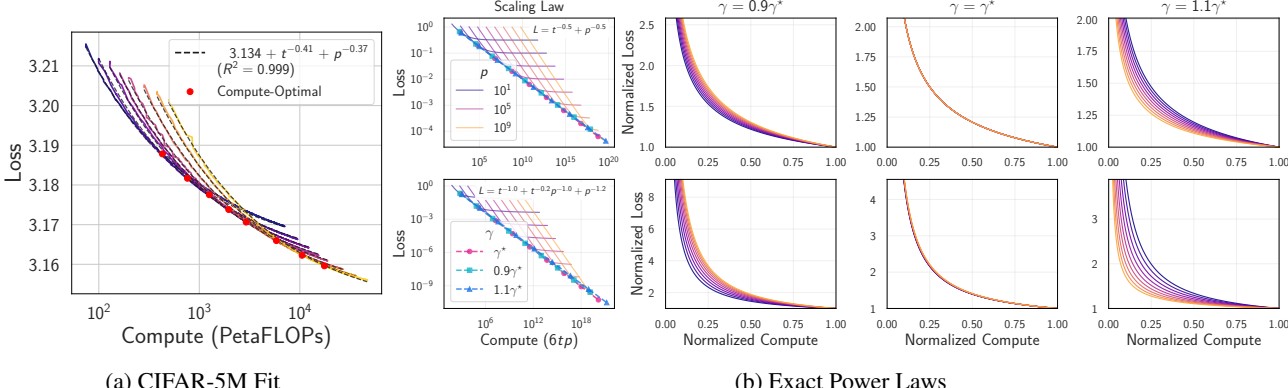

(a) CIFAR-5M Fit
(b) Exact Power Laws

Figure 5: **Scaling collapse from sum of power-law curves.** (**a**) CIFAR-5M expected loss curves (averaged over 5 seed) without learning rate decay agree well with the sum-of-powers-laws fit $L(t, p) = L_0 + t^{-\mu} + p^{-\nu}$ (constant multipliers not shown), a form commonly observed in natural data. We omit the first 1B tokens to avoid fitting the early time transients. (**b**) Simulated exact sum-of-power-laws loss curves show scaling collapse precisely when the data exponent $\gamma$ is the theoretical compute-optimal value $\gamma^\star$. Small variations of $\gamma$ around $\gamma^\star$ lead to nearly negligible worsening in the resulting scaling law but dramatically disrupt the collapse.

# 3. Explaining Loss Curve Scaling Collapse

In this section, we investigate theoretical explanations for the scaling collapse of compute-optimal loss curves and supercollapse. Our analysis starts with a simple observation: the numerator of the collapse deviation $\Delta(x)$ can be decomposed as:

$$\mathbb{V}_{p,\omega}[\ell(x, p, \omega)] = \mathbb{V}_p \mathbb{E}_\omega[\ell(x, p, \omega)] + \mathbb{E}_p \mathbb{V}_\omega[\ell(x, p, \omega)]. \quad (4)$$

The first term corresponds to the variation between different scales $p$ after averaging over all sources of randomness. We will first show how this term can be small:

- In Section 3.1, we prove that for a family of power-law neural scaling laws, compute-optimal loss curves indeed collapse after normalization. We show loss curves in our experiments fall into this family when using a constant learning rate schedule.

- In Section 3.2, we develop a simple theoretical model that successfully predicts the empirical loss curves under various learning rate schedules and explains why they collapse despite deviating from power laws. Given its effectiveness, we believe this model has value for understanding learning rate schedules more broadly.

We then analyze the second term, which captures the loss variance due to random seeds, averaged across model sizes:

- In Section 3.3, we show the same noise model enables us to reason about the noise in the loss curves, and quantitatively predict the variance reduction effect in supercollapse.

Together these findings provide an initial theoretical explanation for supercollapse, and uncover promising directions

for future theoretical work.

## 3.1. Scaling Collapse from Power-Law Scaling

In this section, we consider deterministic models of the loss curves and assume all randomness has been averaged out.

**Power-Law Pareto Frontier is Necessary.** For a family of differentiable loss curves $L(t, p)$, the compute-optimal loss frontier after subtracting $\hat{L}$ must follow a power law for our affine transformation to induce scaling collapse (proof in Appendix E). The key insight is that collapse requires the transformed loss curves to be related by multiplicative scaling, equivalently translation in log-log space, where the frontier must have constant log-log slope since it remains tangent to shifted versions of the same curve. This motivates choosing $\hat{L} = L_0$, which by definition yields the best power-law Pareto frontier. However, a sufficient condition for scaling collapse requires an explicit form of $L(t, p)$.

**Neural Scaling Laws.** Motivated by empirical neural scaling laws in natural data (Hestness et al., 2017; Kaplan et al., 2020; Hoffmann et al., 2022), we consider expected loss curves following a sum-of-power-laws scaling of the form

$$L(t, p) = L_0 + t^{-\mu} + p^{-\nu} \quad (5)$$

for constants $L_0 \geq 0, \mu, \nu > 0$, with potential constant multipliers absorbed via an appropriate choice of units. In Figure 5a, we show the CIFAR-5M loss curves are well-fit by Equation (5) if trained under a constant learning rate schedule (averaged across 5 seeds). We also find decent fits in other datasets in Figure 11.

**Equivalence by Balance of Power Laws.** As before, let $t^\star(p)$ denote the training horizon. We will examine conditions under which $t^\star(p)$ (a) is compute-optimal, and (b) results in scaling collapse. We assume deterministic loss

curves for now and omit the argument $\omega$. To find compute-optimal $t^\star(p)$, we fix $c$ so that $t(p) = c/(6p)$ and minimize the loss $\mathcal{L}(t(p), p) = t(p)^{-\mu} + p^{-\nu}$ with respect to $p$ by setting $0 = \frac{d\mathcal{L}}{dp} = \frac{\partial \mathcal{L}}{\partial t}\frac{dt}{dp} + \frac{\partial \mathcal{L}}{\partial p} = -\mu t^{-\mu-1}(-t/p) - \nu p^{-\nu-1}$

$$\implies \mu t^{-\mu} = \nu p^{-\nu} \qquad (6)$$

which yields $t^\star(p) = r^{-1/\mu} p^{\nu/\mu}$, with $r = \nu/\mu$. Under this scaling, the normalized loss curves are:

$$\ell(x, p) = \frac{(xt^\star)^{-\mu} + p^{-\nu}}{(t^\star)^{-\mu} + p^{-\nu}} = \frac{rx^{-\mu}\bcancel{p^{-\nu}} + \bcancel{p^{-\nu}}}{r\bcancel{p^{-\nu}} + \bcancel{p^{-\nu}}} = \frac{rx^{-\mu} + 1}{r + 1}. \qquad (7)$$

All $p$ dependencies cancel, leaving the final expression independent of $p$ and giving us an exact collapse. Moreover, it is clear that this is the unique choice for $t^\star(p)$ up to a constant multiplier that leads to such cancellation. This agreement is not an accident: compute-optimal scaling requires balancing the derivatives of two power laws, while collapse requires balancing the power laws themselves. For power laws, these two conditions coincide, up to a multiplicative constant.

In Figure 5b, we numerically verify the agreement between collapse and compute-optimal scaling. When the data exponent $\gamma$ deviates from the optimal value $\nu/\mu$, we observe a suboptimal scaling law and no collapse. Note that the absence of an irreducible term in $\ell$ is also necessary. Had we set $\hat{L} = L_0 + E$ for some $E \neq 0$ in Equation (1), we would instead have $\ell(x, p) = \frac{(xt^\star)^{-\mu} + p^{-\nu} + E}{(t^\star)^{-\mu} + p^{-\nu} + E}$, where no $t^\star(p)$ can leave the numerator and denominator homogeneous in $p$.

In Appendix F, we study the more general form

$$L(t, p) = L_0 + \sum_{i=1}^{m} a_i t^{-\mu_i} p^{-\nu_i}, \qquad (8)$$

which naturally arises in theoretical models of neural scaling laws (Paquette et al., 2024b; Bordelon et al., 2024a;b), and show that compute-optimality implies scaling collapse by balancing the two dominant terms, though with $m > 2$ the collapse is only exact asymptotically.

Together with the close empirical fit in Figure 5a, this analysis explains scaling collapse in the constant learning rate setting; however Equation (5) fails to fit the empirical loss curves with most learning rate schedules, as varying the learning rate can modulate the loss curve in quite arbitrary ways, clearly shown in Figure 1d. Why, then, does the collapse transfer to other schedules?

## 3.2. Universality of Learning Rate Schedules

To understand why scaling collapse is robust across learning rate schedules, we develop a quantitative model for how learning rate schedules affect the loss curves. While an exact theoretical model seems out of reach for the realistic training setup, we show that a simple model based on quadratic loss analysis proves surprisingly effective. Under this model, we demonstrate that although learning rate schedules deform the loss curves in a schedule-dependent way, the deformation is approximately independent of $p$. We consider stochastic effects that depend on the random seed $\omega$, but omit $\omega$ as an explicit argument for brevity and use bar to denote expectation over $\omega$.

### 3.2.1. A SIMPLE MODEL FOR LR SCHEDULES

Let $w(t)$ and $L(w(t))$ denote the parameters and loss at step $t$, we can model the dynamics of full-batch gradient descent under a small learning rate $\eta(t)$ with a gradient flow $\frac{dw}{dt} = -\eta(t)\nabla L(w(t))$. To model stochastic effects, a noise term is added to the gradient, leading to the SDE $\frac{dw}{dt} = -\eta(t)\big(\nabla L(w) + \Sigma^{1/2}(w)\xi(t)\big)$ (Li et al., 2017; Malladi et al., 2022), where the *mini-batch* gradient noise $\Sigma^{1/2}(w)\xi(t)$ satisfies $\mathbb{E}[\xi(t)\xi(t')] = \delta(t - t')I$, and we allow its covariance (which depends on batch size) $\Sigma(w)$ to be a function of the parameters. Prior works have used the SDE model or discrete variants to study learning rate schedules in analytically tractable problems (Zhang et al., 2019; d'Ascoli et al., 2022; Wen et al., 2024), but we will show it can make surprisingly accurate predictions in real models. We work in *gradient flow time* $\tau(t) = \int_0^t \eta(s)ds$, where

$$\frac{dw}{d\tau} = -\Big(\nabla L(w) + \Sigma^{1/2}(w)\xi(\tau)\Big), \qquad (9)$$

and $\mathbb{E}[\xi(\tau)\xi(\tau')] = \delta(t - t')I = \eta(\tau)\delta(\tau - \tau')I$. We overload the notation and use $\eta(\tau)$, $w(\tau)$, and $L(\tau)$ to denote the evolution of these quantities in gradient flow time.

**Quadratic Loss.** For the moment, let us suppose the loss function is quadratic $L(w) = \frac{1}{2}w^\top H w$, where we assume the minimum is at the origin without loss of generality. Then $\nabla L(w) = Hw$ and standard calculation shows

$$w(\tau) = e^{-H\tau}w(0) - \int_0^\tau ds\, e^{-H(\tau-s)}\Sigma^{1/2}(w(s))\xi(s). \qquad (10)$$

Letting $\bar{\Sigma}(s) = \mathbb{E}[\Sigma(w(s))]$, the expected loss is then

$$\bar{L}(\tau) = \underbrace{\frac{1}{2}\mathbb{E}\big[\|e^{-H\tau}w(0)\|_H^2\big]}_{\mathscr{F}(\tau)}$$

$$+ \underbrace{\frac{1}{2}\int_0^\tau ds\, \eta(s)\,\mathrm{Tr}\Big(He^{-2H(\tau-s)}\bar{\Sigma}(s)\Big)}_{\mathcal{E}(\tau)}. \qquad (11)$$

The first term $\mathscr{F}(\tau)$ is the forcing function, equal to the expected loss curve in the noiseless limit $\eta\Sigma \to 0$ and is

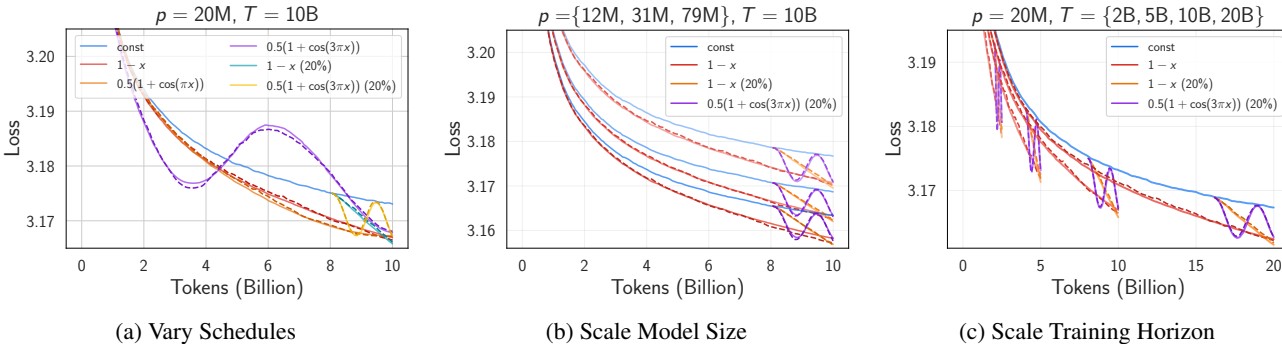

Figure 6: **A simple model predicts Transformer loss curves trained across learning rate schedules, model sizes $p$, and training horizons $T$ on CIFAR-5M.** Dashed curves show the predicted loss according to Equation (18), with $\alpha = 0.21$, which closely match with the true curves in solid. Each curve is smoothed with an exponential moving average with half-life equal to $1\%$ of total steps.

independent of the learning rate schedule. The second term $\mathcal{E}(\tau)$ is the excess loss due to SGD noise, which is a sum of exponential moving averages (up to normalization) of the gradient variance scaled by the learning rate over each eigenmode. Substituting in the specific forms for $\Sigma$ recovers the convolutional Volterra equation for linear regression analyzed in Paquette et al. (2021; 2024a), or the noisy quadratic model in Zhang et al. (2019) for small learning rates.

If $\eta\bar{\Sigma}$ varies slowly compared to the timescale of the exponential moving average, we can make the approximation $\eta(s)\bar{\Sigma}(s) \approx \eta(\tau)\bar{\Sigma}(\tau)$ inside the integrand, giving us:

$$\mathcal{E}(\tau) \approx \frac{1}{2}\eta(\tau)\operatorname{Tr}\left(\bar{\Sigma}(\tau)H\int_0^\tau ds\, e^{-2H(\tau-s)}\right) \quad (12)$$

$$= \frac{1}{4}\eta(\tau)\operatorname{Tr}\left(\bar{\Sigma}(\tau)\left(1 - e^{-2H\tau}\right)\right). \quad (13)$$

For large $\tau$ the expected loss is then approximately

$$\bar{L}(\tau) \approx \mathscr{F}(\tau) + \frac{1}{4}\eta(\tau)\operatorname{Tr}\left(\bar{\Sigma}(\tau)\right). \quad (14)$$

Given access to $\operatorname{Tr}\left(\bar{\Sigma}(\tau)\right)$, we can derive a prediction for how the loss changes as we change the learning rate schedule without knowing $\mathscr{F}$.

**General Case.** In Appendix G, we discuss how this analysis can be generalized to more realistic setups. For general loss functions, we show via perturbation theory that, to first order in $\eta\bar{\Sigma}$, one can make similar approximations to derive Equation (14) given an additional assumption that the Hessian is slowly varying, and with the forcing function $\mathscr{F}(\tau)$ no longer admitting a quadratic form. We also show in Appendix G that $\Sigma$ should be the *preconditioned* gradient covariance when using adaptive optimizers. We absorb the layerwise, width-dependent learning rates in $\mu$P into the preconditioner, similar to Noci et al. (2024), so $\eta(t) \in [0, 1]$ reflects only the schedule.

### 3.2.2. PREDICTING LOSS CURVES ACROSS SCHEDULES

We apply this simple model to predict empirical loss curves in the CIFAR-5M experiments. We measure the trace of the preconditioned gradient covariance on a fixed set of 2M tokens (see Appendix A for experiment details).

Let $\bar{L}, \eta, \bar{\Sigma}$ be a given reference trajectory and $\bar{L}' = \bar{L} + \delta\bar{L}, \eta' = \eta + \delta\eta, \bar{\Sigma}' = \bar{\Sigma} + \delta\bar{\Sigma}$ be the target trajectory, Equation (14) allows us to predict the target loss via

$$\delta\bar{L}(\tau) \approx \frac{1}{4}\operatorname{Tr}\left[\delta\left(\eta(\tau)\bar{\Sigma}(\tau)\right)\right], \quad (15)$$

where $\delta\left(\eta(\tau)\bar{\Sigma}(\tau)\right) := \eta'(\tau)\bar{\Sigma}'(\tau) - \eta(\tau)\bar{\Sigma}(\tau)$. We use a constant learning rate for the reference trajectories and various schedules sharing the same peak learning rate for the target. Decomposing $\delta(\eta\Sigma) = \delta\eta\Sigma' + \eta\delta\Sigma$, we find the first term is typically 3 to 10 times larger than the second as the learning rate decays, which can be attributed to how learning rate interacts with curvature (Figure 13). In Figure 6, we only keep the first term, and predict the target loss as

$$L'(\tau) \approx L(\tau) + \alpha\,\delta\eta(\tau)\operatorname{Tr}(\Sigma'(\tau)), \quad (16)$$

where $\alpha$ is a shared hyperparameter. We find a single $\alpha = 0.21$ fits the target loss curves surprisingly well across schedules, model sizes, and training horizons. In Appendix I, we show even better fits for MLPs, though puzzlingly, including the second term can lead to worse fits.

Recent works proposed more complex functional forms for how learning rate schedules affect loss curves, derived primarily from empirical observations (Tissue et al., 2024; Luo et al., 2025). The accuracy of our simple model suggests it captures the essential dynamics, and crucially, the correct scaling of the excess loss through $\operatorname{Tr}(\Sigma')$ so that a single $\alpha$ is predictive across model sizes, schedules, and training horizons. Notably, Luo et al. (2025) experimented with a similar form to Equation (16) but with a constant $\Sigma'$, which likely explains the reduced effectiveness they observed.

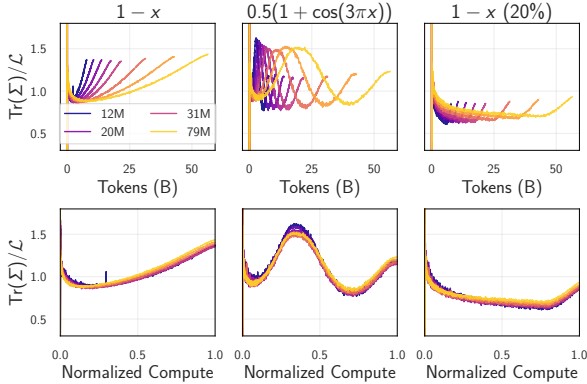

Figure 7: **Universality of gradient noise on CIFAR-5M.** Fixing a learning rate schedule, the ratio $\mathrm{Tr}(\Sigma)/\mathcal{L}$ is approximately a function of normalized compute alone, independent of model size. We show similar results with MLP regression in Figure 14.

### 3.2.3. UNIVERSAL SCALING OF GRADIENT NOISE

For typical loss functions, the gradient covariance can be related to the loss itself. For example, in noiseless high-dimensional linear regression with Gaussian features drawn from $\mathcal{N}(0, K)$, we have $\mathrm{Tr}(\Sigma) \approx 2\mathcal{L} \mathrm{Tr}(K)$ (Paquette et al., 2021), an intuitive result since the gradient scales with both the prediction error and the input. For non-linear regression, $K$ should be taken to be the time-varying Gauss-Newton matrix for a first approximation. In this case, $\mathrm{Tr}(K)$ is known to depend strongly with the learning rate (Agarwala & Pennington, 2024), but we expect weak dependence on model size given our models are trained with $\mu$P (see Noci et al. (2024) for evidence that curvature statistics depend weakly on model size in $\mu$P). Since the schedule is a function of the normalized compute $x = t/t^\star$ alone, we hypothesize there exists a schedule-dependent function $h(x)$ such that

$$\mathrm{Tr}(\Sigma(xt^\star(p)))/\mathcal{L}(xt^\star(p)) \approx h(x), \quad (17)$$

which we verify in the regression (Figure 14) and next-token prediction experiments (Figure 7).

Combining Equation (16) and Equation (17) and making $p$-dependence explicit:

$$\bar{\mathcal{L}}'(\tau, p) \approx \bar{\mathcal{L}}(\tau, p)(1 - \alpha h(x)\delta\eta(\tau, p))^{-1}, \quad (18)$$

where $x$ is the normalized compute at gradient flow time $\tau$. We leave to future work an explanation of why this relation appears to hold for cross-entropy loss despite the presence of non-negligible irreducible loss, as this setting is analogous to regression with label noise, where the gradient covariance should scale with the total loss rather than just the reducible component, i.e., $\mathrm{Tr}(\Sigma) \approx 2L \mathrm{Tr}(K)$.

**Scaling Collapse Across Schedules.** Combining our insights so far, we can now understand why collapse happens across schedules. Let $\bar{\ell}(x, p)$ and $\bar{\ell}'(x, p)$ be the expected normalized loss curves under two schedules $S$ and $S'$. Let $y(x)$ map the normalized compute under $S'$ to the normalized compute under $S$ at matching gradient flow time, where $y$ is independent of $p$ for schedules defined in terms of the normalized compute. Let $\delta\hat{\eta}(x) = \delta\eta(xt^\star(p), p)$ be the difference between the two schedules. Assuming small relative fluctuations ($\mathbb{E}[\mathcal{L}(x)/\mathcal{L}(y)] \approx \mathbb{E}[\mathcal{L}(x)]/\mathbb{E}[\mathcal{L}(y)]$), we have:

$$\bar{\ell}'(x, p) \approx \frac{\bar{\mathcal{L}}'(xt^\star(p), p)}{\bar{\mathcal{L}}'(t^\star(p), p)} \quad (19)$$

$$= \frac{\bar{\mathcal{L}}(y(x)t^\star(p), p)(1 - \alpha h(x)\delta\hat{\eta}(x))^{-1}}{\bar{\mathcal{L}}(y(1)t^\star(p), p)(1 - \alpha h(1)\delta\hat{\eta}(1))^{-1}} \quad (20)$$

$$= \bar{\ell}(y(x), p) \underbrace{\frac{1 - \alpha h(1)\delta\hat{\eta}(1)}{1 - \alpha h(x)\delta\hat{\eta}(x)}}_{\text{independent of } p}, \quad (21)$$

which shows that, in expectation, collapse under one schedule (e.g. constant) implies collapse under any other schedule, provided we take Equation (18) to be exact. Since collapse under a constant learning rate can be attributed to the sum-of-power-laws scaling law, this result helps explain why we also observe collapse in other schedules.

This analysis also suggests that collapse can serve as a filter for identifying interventions that yield scalable improvements: those that multiplicatively shift the reducible loss curve by the same factor across all model sizes.

### 3.3. Supercollapse as Variance Reduction

Lastly, we turn to understanding the "super" in supercollapse: why does learning rate decay significantly improve the collapse, to the extent that the collapse deviation $\Delta(x)$ drops below the per-model noise floor $\sigma(x, p)$ for a substantial fraction of training? Again, the simple quadratic model provides quantitative insights into this phenomenon.

Recall $\Delta(x) = \frac{\mathbb{V}_{p,\omega}[\ell(x,p,\omega)]^{1/2}}{\mathbb{E}_{p,\omega}[\ell(x,p,\omega)]}$, and the decomposition:

$$\mathbb{V}_{p,\omega}[\ell(x,p)] = \mathbb{E}_p\mathbb{V}_\omega[\ell(x,p)] + \mathbb{V}_p\mathbb{E}_\omega[\ell(x,p)]. \quad (22)$$

The first term measures variance due to the seed alone, averaged over model sizes, while the second term measures variance due to varying the model size, having averaged over the seeds first. Since we observed that variations in the normalized curves primarily arise from seed-to-seed fluctuations rather than model-to-model differences (Section 2.4) under a constant schedule, and switching to other schedules does not significantly increase the model-to-model differences (Section 3.2), we will assume the first term $\mathbb{E}_p\mathbb{V}_\omega[\ell(x,p)]$ dominates, which implies $\Delta^2(x) \approx \mathbb{E}_p\tilde{\Delta}^2(x,p)$, where $\tilde{\Delta}^2(x,p) := \mathbb{V}_\omega[\ell(x,p)]/\bar{\ell}^2(x,p)$ is the squared *per-model* collapse deviation.

To simplify notation, we temporarily omit $p$-dependence and write $\ell$ in terms of $t$ instead of $x$. Letting $\mathcal{L}(t) =$

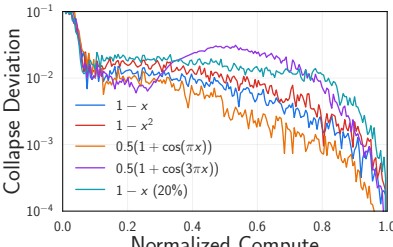 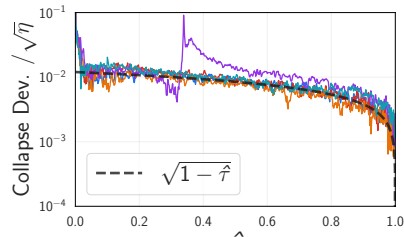 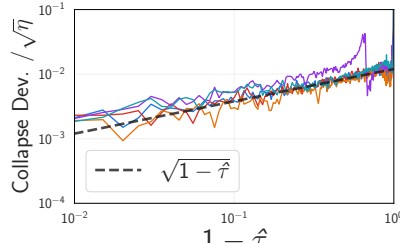

Figure 8: **A quantitative explanation of how learning rate decay leads to supercollapse.** Across schedules on CIFAR-5M, collapse deviation at normalized gradient flow time $\hat{\tau}$ follows the predicted $\sqrt{\eta(1-\hat{\tau})}$ scaling, capturing the noise accumulated between that point and end of training. A schedule that decays faster has a smaller $\eta$ and $(1-\hat{\tau})$ at a fixed normalized compute or training step.

$\bar{\mathcal{L}}(t)(1+\psi(t))$, where $\psi$ is the relative fluctuation, we have $\ell(t) = \frac{\bar{\mathcal{L}}(t)(1+\psi(t))}{\bar{\mathcal{L}}(t^\star)(1+\psi(t^\star))} \approx \bar{\ell}(t)(1+\psi(t)-\psi(t^\star))$, assuming $\psi \ll 1$. Therefore,

$$\tilde{\Delta}^2(t) \approx \mathbb{E}\big[(\psi(t)-\psi(t^\star))^2\big] \qquad (23)$$

We see that what controls the relative variance in $\ell(t)$ is not $\psi(t)$ but the difference $\psi(t) - \psi(t^\star)$, which roughly captures the amount of optimization noise accumulated *between* time $t$ and time $t^\star$. Since the optimization noise per step scales with the instantaneous learning rate, decaying the learning rate over time will precisely serve to decrease the variance in $\ell$. By contrast, the squared per-model noise floor $\sigma^2(t)$ is simply $\mathbb{E}[\psi^2(t)]$, which captures the total cumulative optimization noise. Importantly, had we normalized by the expected rather than the empirical final loss in $\ell$, $\tilde{\Delta}(t)$ would reduce to $\sigma(t)$. Normalizing by the stochastic final loss is essential for supercollapse, where it acts as a control-variate (Glasserman, 2004), leveraging the strong time-correlation of stochastic fluctuations along the optimization trajectory to cancel much of the shared noise and thereby sharply reduce the variance of the collapsed curve.

Quantitatively, we can estimate $\tilde{\Delta}$ under the quadratic model in Section 3.2. Let $\Delta w(\tau)$ and $\Delta\mathcal{L}(\tau)$ be the fluctuations of the parameters and loss from their means. We have $\Delta w(\tau) = \int_0^\tau ds\, e^{-H(\tau-s)}\Sigma^{1/2}(s)\xi(s)$, and $\Delta\mathcal{L}(\tau) = g(\tau)^\top \Delta w(\tau)$ to first order in $\Delta w(\tau)$, where $g(\tau)$ is the expected gradient. Close to the end of training, for $\tau = \tau^\star - \delta\tau$ where $\tau^\star$ is the final gradient flow time and $\delta\tau > 0$ is small, direct calculation shows (Appendix H)

$$\tilde{\Delta}^2(\tau) = \bar{\mathcal{L}}^{-2}(\tau)g(\tau)^\top \eta(\tau)\bar{\Sigma}(\tau)g(\tau)\delta\tau + O(\delta\tau^2), \quad (24)$$

In linear regression, $\Sigma \propto \mathcal{L}$ and $g^\top g \propto \mathcal{L}$, so we estimate $\tilde{\Delta}^2(\tau) \propto \eta(\tau)\delta\tau$ to leading order. Since this relation holds for each model size $p$, we predict $\Delta^2(\hat{\tau}) \approx \mathbb{E}_p\tilde{\Delta}^2(\hat{\tau}) \propto \eta(\hat{\tau})(1-\hat{\tau})$, where $\hat{\tau} = \tau/\tau^\star$ denotes the normalized gradient flow time. Figure 8 shows this form fits the observations well, with $\Delta(\hat{\tau})/\sqrt{\eta(\hat{\tau})}$ approximately following the same $\sqrt{1-\hat{\tau}}$ scaling across many schedules, quantitatively explaining how learning rate decay leads to supercollapse.

## 4. Discussion

Scale has enabled remarkable progress in machine learning, but a thorough scientific understanding of scaling remains elusive. Key open questions include identifying robust principles that guide general hyperparameter transfer and characterizing scaling limits under realistic scaling ladders. Our discovery of supercollapse provides empirical evidence that a model-size and data joint scaling limit *generically* exists in the compute-optimal regime, and that the scale-invariance of the training dynamics revealed by the collapse can diagnose proper hyperparameter configuration. We believe further investigation of these phenomena holds great potential for advancing the science of scaling.

We see many exciting extensions to this work. Empirically, our small-scale experiments provide a proof-of-concept. While small-scale proxies capture certain behaviors in larger systems (Wortsman et al., 2023), validating at larger scales and with practical scaling ladders, where width, depth, batch size, and weight decay are co-scaled (McCandlish et al., 2018; Wang & Aitchison, 2024; Dey et al., 2025; Bergsma et al., 2025), is important and may yield new insights into optimal scaling and hyperparameter transfer. Scaling collapse beyond the form we studied here can be a general tool to study other scaling relations (Tamai et al., 2023).

While we have identified the key ingredients underlying supercollapse—power-law scaling and learning rate-dependent noise dynamics—our analysis relies on multiple approximations and takes power-law scaling as given, suggesting deeper theoretical principles may be at work. Taking collapse as a starting point instead may provide an alternative route to understanding scaling laws, analogous to how in physics the renormalization group was developed to provide a unified set of principles explaining both universality and its associated power laws (Wilson, 1971). Finally, it would be interesting to understand why our simple noise model predicts the impact of learning rate schedules on real models so effectively, compare it with alternative models such as Schaipp et al. (2025), and to test its predictive power for optimizing schedules, learning rates, and training horizons.

## Acknowledgements

We thank Courtney Paquette and Zixi Chen for helpful comments on an earlier version of this paper. SQ was supported by Google's TPU Research Cloud (TRC) program: https://sites.research.google/trc/.

## Contribution Statement

SQ designed and conducted the majority of experiments, led the theory development, and wrote the paper. LX initially observed supercollapse, contributed to theory, experimental design, and writing the paper. AGW advised SQ and edited the paper. JP contributed to theory, experimental design, and writing the paper. AA managed the research project, proved some theorems, guided the theory development, contributed to experimental design, and helped write the paper.

## Impact Statement

This paper presents work whose goal is to advance the field of Machine Learning. There are many potential societal consequences of our work, none which we feel must be specifically highlighted here.

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

# A. Experiment Details

**Transformer Architecture.** We use GeLU activations (Hendrycks & Gimpel, 2016), RMSNorm (Zhang & Sennrich, 2019), and learned positional embeddings. We untie the embedding matrix from the output head and do not use bias anywhere. The readout layer is always zero-initialized, as suggested by Yang et al. (2021) . We denote the embedding dimension with $D$. We set the intermediate dimension in the feedforward layers to $D$ instead of the usual $4D$, which enables us to explore larger widths more efficiently. The head dimension is set to 64.

**CIFAR-5M.** We use the CIFAR-5M dataset (Nakkiran et al., 2020) of 6 million CIFAR-like images. We convert the $32 \times 32 \times 3$ images to greyscale and flatten them into sequences of length 1024. The model autoregressively predicts the pixel intensities in raster-scan order. The vocabulary is the set of pixel intensities $\{0, \ldots, 255\}$. Following $\mu$P we parameterize the learning rate for each weight matrix as $\eta = \eta_{\text{base}}/D$ where $d$ is the model dimension, except for the embedding matrix which has $\eta = \eta_{\text{base}}$. We use a parameter multiplier $a$ on the embedding matrix. We use $\eta_{\text{base}} = 4$ and $a = 0.1$ as they led to good performance in our early experiments. We initialize the embedding matrix as $W_{ij}^{\text{emb}} \sim \mathcal{N}(0, 1)$, the output head as $W^{\text{head}} = 0$, all other non-readout matrices $W$ as $W_{ij} \sim \mathcal{N}(0, 1/D)$. These hyperparameters were determined with a small amount of tuning in early experiments. We use a batch size of 256 images. We use a linear warmup for 1000 steps.

For the experiments in Section 3.2.2, we used a slightly different setup due to switching to a new codebase in the middle of the research project. We use $\mu$P where the base embedding dimension is 128 and base learning rate is 0.01. We initialize the embedding matrix with a standard deviation (std) of 0.1 and multiply its learning rate by 10 relative to the base learning rate. The output projection of the feedforward and attention layers are zero-initialized. All other non-readout matrices are initialized with std $1/\sqrt{D}$. We use a batch size of 65536 tokens or 64 images. We use a linear warmup for 10M tokens.

**Chess.** We run our experiments on the Lichess dataset available on Hugging Face at `https://huggingface.co/datasets/Lichess/standard-chess-games`. We used character-level tokenization and a context length of 128. We use $\mu$P where the base embedding dimension is 128 and base learning rate is 0.01. We initialize the embedding matrix with std 0.1 and multiply its learning rate by 10 relative to the base learning rate. The output projection of the feedforward and attention layers are zero-initialized. All other matrices are initialized with std $1/\sqrt{D}$. We use a batch size of 65536 tokens. We use a linear warmup for 10M tokens.

**MLP Experiments.** Our MLP architecture is identical to the transformer with attention layers removed and the token and position embedding layers replaced by a linear layer. We use $\mu$P where the base width is 128 and base learning rate is 0.001. The output projection of the feedforward layers are zero-initialized. All non-readout matrices are initialized with std $1/\sqrt{D}$. We use a batch size of 4096 examples. We do not use warmup.

The target function is defined as $\phi(x) = \sum_{i=1}^{M} w_i \sqrt{2} \cos(2\pi k_i^\top x + b_i)$, with $x \in \mathbb{R}^8, w_i \sim \mathcal{N}(0, 1), b_i \sim \frac{\pi}{2}\text{Bernoulli}(0.5), k_i = \text{round}(s_i v_i)$ where $s_i$ is a scalar sampled from a power law with support $[1, \infty)$ and exponent $-2$, $v_i$ is a random unit vector, and round rounds to the nearest point in $\mathbb{Z}^8$. During training, $x$ is sampled uniformly from $[-0.5, 0.5]^8$, making the Fourier features orthonormal over the data distribution. We suspect the details here are not necessary for generating power-law scaling laws beyond the power-law spectrum.

For Figure 4a, the constant learning rate scaling ladder matches the $\mu$P learning rate at $D = 384$ (smallest model).

**Measuring Gradient Covariance Trace.** As mentioned in Section 3.2.1, we use the preconditioned gradient covariance $\tilde{\Sigma}$ instead of the raw gradient covariance due to the use of Adam (and $\mu$P). $\tilde{\Sigma}$ is defined as $P^{-1/2}\Sigma P^{-1/2}$ where $\Sigma$ is the raw (mini-batch) gradient covariance, and $P$ is the preconditioner. See an explanation for this definition in Appendix G. Since $\mu$P uses layerwise learning rate, the preconditioner is defined as $P^{-1} = \text{diag}\left(\frac{\eta_0}{\sqrt{v^2}+\epsilon}\right)$, where $\eta_0$ is the vector of per-parameter learning rate (peak learning rate, before applying the schedule), $v^2$ is the Adam second-moment, and $\epsilon$ is the Adam $\epsilon$. As the peak learning rate is absorbed into the preconditioner, any occurrence of the instantaneous learning rate $\eta(t)$ or $\eta(\tau)$ in Section 3.2.2 reflects only the schedule and takes on values in $[0, 1]$. This definition mirrors what is done in Noci et al. (2024); Cohen et al. (2024).

**Training for More than One Epoch.** The CIFAR-5M dataset, tokenized as $32 \times 32$ greyscale images, has about 5B tokens. Therefore, most models needed to be trained for more than 1 epoch to reach the compute-optimal training horizon. Up to the scales we tested, we did not observe a significant difference between the train and test loss. The chess dataset has about 20B tokens, which also led to data reuse for some models, but did not lead to significant overfitting. As we only processed a subset of the full Lichess dataset, this can be avoided by processing a larger subset if desired.

**On Random Seeds.** In the CIFAR-5M experiments in Figure 1, the random seed controls both the initialization and data ordering. In the other experiments (Transformer on chess and MLP regression), the random seed only controls the initialization while data ordering is held fixed, as is often done in practice. Fixing the data ordering (no shuffling) had the advantage of speeding up data loading. We found that supercollapse occurs regardless of whether seeds affect data ordering. This makes sense: even with fixed ordering, different model sizes process different data due to varying training horizons. More fundamentally, supercollapse should be robust to which training components are randomized, as the variance reduction arises from strong noise correlations along individual trajectories rather than specific noise sources (Section 3.3).

## B. Scaling Collapse Across Transformer Depths

For scaling depth, we additionally apply a branch multiplier of $3/\mathrm{depth}$ on the output of every feedforward and attention layer, as suggested by Bordelon et al. (2024c). We find a decent degree of collapse in Figure 9 when training on chess data. There is a small shift in the normalized curves, though we are unsure if it is simply a finite-size effect.

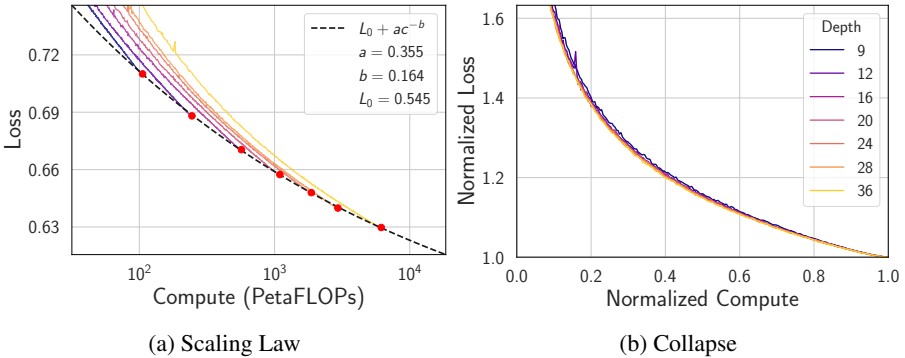

(a) Scaling Law         (b) Collapse

Figure 9: **Depthwise scaling collapse for transformers trained on chess.**

## C. Estimating Compute-Optimal Training Horizon

To estimate the optimal compute for training each model, we perform the following steps in each experiment:

- We trained each model *without* learning rate decay but keeping the initial warmup. We chose a large enough number of steps so that the largest model could reach the compute-optimal loss frontier. We average the loss curves from 5 seeds.

- We numerically computed the compute-loss Pareto frontier to obtain an estimate of $c^\star(p)$ - the optimal compute for each model size $p$. We use logarithmically spaced points for $c$ and find the $p$ that achieves the best loss given $c$ training FLOPs.

- We fit a power law $c^\star(p) = \kappa p^{1+\gamma}$ where $\kappa$ and $\gamma$ are fit parameters. The optimal number of training tokens is then $t^\star(p) = c^\star(p)/(6p)$, which scales as $p^\gamma$. We remove outliers in this fit by dropping points from the smallest and largest model.

Using a constant learning rate schedule allows us to measure the loss of one model at different token budgets with a single training run, an approach also used in McLeish et al. (2025), rather than one training run per token budget as origianlly done in Hoffmann et al. (2022). Figure 10 illustrates this procedure.

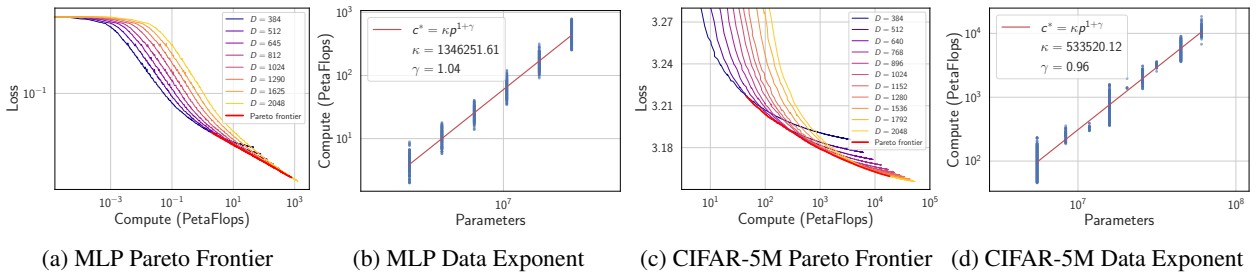

| (a) MLP Pareto Frontier | (b) MLP Data Exponent | (c) CIFAR-5M Pareto Frontier | (d) CIFAR-5M Data Exponent |

Figure 10: **Estimating compute-optimal data exponent in the MLP regression and Transformer CIFAR-5M experiments.**

## D. Universality and Scaling Collapse in Other Sciences

The simplest versions of collapse come from statistics and probability, where entire *distributions* of random variables show universal behavior between systems of different types and scales. The most well known is the central limit theorem which predicts a universal Gaussian form for the sums of random variables with appropriately bounded moments (and Levy distributions for heavy tailed distributions). In random matrix theory, there are similar effects when studying the limiting empirical distribution of spectra, most famously the Marcenko-Pastur distribution of Wishart matrices (Marchenko & Pastur, 1967). In all of these examples, showing universal behavior of a single moment is analogous to showing predictability of the Pareto frontier in our work, while showing the universality of the whole distribution is analogous to our statements about the entire loss curves (where e.g. the CDFs of different problems are converging).

Scaling collapse is ubiquitous in physics as well. There are again distributional collapses like the famed Maxwell Boltzmann distribution first used to describe idealized gasses, but also functional relationships like the universal magnetization-temperature curves of near-critical Ising lattices of different sizes (Binder, 1981). In the Ising example, changes to the lattice topology can change the magnetization-temperature curves, similar to how different datasets, architecture, and training algorithms lead to different universal curves in our study. A more general theory unifying and explaining the existence of universality in physics arises from the renormalization group (Wilson, 1971).

More recently, scaling collapse has been used to describe dynamical systems in biological contexts. Advances in genomics have led to the rise of experimental microbial evolution with rapid timecourse data (Levy et al., 2015; Venkataram et al., 2016). Analysis of this data relies on quantitative modeling of evolutionary dynamics. Most of these models show universal scaling dynamics, in situations from rapid evolutions of diverse populations (Fisher, 2013), populations evolving under changing fitness conditions (Agarwala & Fisher, 2019), and populations expanding in space (Hallatschek & Fisher, 2014). In these settings the timecourse of key observables can be described with dynamical curves which can be rescaled to universal forms with transformations depending on population size, mutation frequency, and statistics of the fitness landscape. In ecology, the ubiquity of fine-scale diversity and the seemingly universal, power-law nature of species rank-abundance curves (Rosen et al., 2015; Ser-Giacomi et al., 2018) can be explained using dynamical models which themselves show universal scaling behavior over ecosystems of different sizes (Pearce et al., 2020).

Scaling collapse has been used to study certain scaling relations in machine learning. Kaplan et al. (2020) identified universal scaling of overfitting, showing a collapse of the rescaled excess loss vs rescaled parameter count across dataset sizes. Tamai et al. (2023) used scaling collapse to establish universal scaling laws in the forward signal propagation dynamics of MLPs near the order-to-chaos transition.

## E. Power-Law Pareto Frontier is Necessary for Collapse

Recall $t^\star(p)$ is the optimal training horizon for model size $p$, i.e. $L(t^\star(p), p) = \min_{t', p':t'p'=t^\star(p)p} L(t', p')$. Let $c^\star(p) = 6t^\star(p)p$ be the optimal compute for $p$. In what follows, rather than writing $L(t, p)$, we will find it convenient to express the loss curves in terms of compute and model size. Letting $L(c, p)$ be the loss curves expressed this way, we have the following theorem:

**Theorem E.1.** *Let $\mathcal{L}(c, p)$ be $C^1$ for all positive $c, p$ and let $c^\star(p)$ denote the compute-optimal budget for model size $p$.*

Write $\mathcal{L}(c,p) = L(c,p) - \hat{L}$ for some offset $\hat{L}$ (e.g., $\hat{L} = L_0$ the irreducible loss). Define the normalized loss curve

$$\ell(x,p) = \frac{\mathcal{L}(xc^\star(p), p)}{\mathcal{L}(c^\star(p), p)}, \quad x \in [0,1]. \tag{25}$$

*Then,*

**1. Necessity.** *If $\ell$ is independent of $p$ (collapse), then the Pareto frontier of $\{\mathcal{L}(c,p)\}_{c,p}$*

$$\mathcal{L}^\star(c) := \min_p \mathcal{L}(c,p) = \mathcal{L}(c^\star(p), p) \tag{26}$$

*is a power law $\mathcal{L}^\star(c) = ac^{-\delta}$ for some constants $a, \delta$.*

**2. Sufficiency at first order.** *Conversely, suppose $\mathcal{L}^\star(c) = ac^{-\delta}$, then*

$$\left. \frac{d\ell(x,p)}{dx} \right|_{x=1} = -\delta, \tag{27}$$

*independent of $p$. Hence all curves share the same first-order behavior around $x = 1$, i.e., they collapse to first order around $x = 1$.*

*Proof.* First, we have the following identity for log-derivatives for a general differentiable function $u(v)$

$$\frac{d\log u}{d\log v} = \frac{v}{u}\frac{du}{dv}. \tag{28}$$

Applying this to our normalized curve from Equation (25) and using the chain rule:

$$\left. \frac{d\ell(x,p)}{dx} \right|_{x=1} = \frac{c^\star}{\mathcal{L}(c^\star, p)} \left. \frac{\partial \mathcal{L}(c,p)}{\partial c} \right|_{c=c^\star} \tag{29}$$

$$= \left. \frac{\partial \log \mathcal{L}(c,p)}{\partial \log c} \right|_{c=c^\star}, \tag{30}$$

where the second equality follows from Equation (28).

**Necessity.** If $\ell$ is independent of $p$ (collapse), then $\left. \frac{d\ell(x,p)}{dx} \right|_{x=1}$ is the same for all $p$. Set this common value to $-\delta$. By Equation (30),

$$\left. \frac{\partial \log \mathcal{L}(c,p)}{\partial \log c} \right|_{c=c^\star(p)} = -\delta \quad \text{for every } p. \tag{31}$$

Since $(c^\star(p), p)$ lies on the Pareto frontier and $\mathcal{L}$ is $C^1$, the envelope theorem states

$$\left. \frac{d\mathcal{L}^\star(c)}{dc} \right|_{c=c^\star(p)} = \left. \frac{\partial \mathcal{L}(c,p)}{\partial c} \right|_{c=c^\star(p)}, \tag{32}$$

i.e. the loss curve is tangent to the Pareto frontier at the compute-optimal point. Applying Equation (28) to the frontier $\mathcal{L}^\star(c)$:

$$\left. \frac{d\log \mathcal{L}^\star(c)}{d\log c} \right|_{c=c^\star(p)} = \frac{c^\star(p)}{\mathcal{L}^\star(c^\star(p))} \left. \frac{d\mathcal{L}^\star(c)}{dc} \right|_{c=c^\star(p)} \tag{33}$$

$$= \frac{c^\star(p)}{\mathcal{L}(c^\star(p), p)} \left. \frac{\partial \mathcal{L}(c,p)}{\partial c} \right|_{c=c^\star(p)} \tag{34}$$

$$= \left. \frac{\partial \log \mathcal{L}(c,p)}{\partial \log c} \right|_{c=c^\star(p)} \tag{35}$$

$$= -\delta, \tag{36}$$

where we used Equation (32) and Equation (31). This means the log-log slope of the frontier is constant, which means it is a power law $\mathcal{L}^\star(c) = ac^{-\delta}$.

**Sufficiency at first order.** Assume $\mathcal{L}^\star(c) = ac^{-\delta}$. For any model size $p$, the envelope theorem gives the tangency condition at $c = c^\star(p)$:

$$\left.\frac{\partial \mathcal{L}(c, p)}{\partial c}\right|_{c=c^\star} = \left.\frac{d\mathcal{L}^\star(c)}{dc}\right|_{c=c^\star} = -\delta a (c^\star)^{-\delta-1}. \tag{37}$$

Applying Equation (30):

$$\left.\frac{d\ell(x, p)}{dx}\right|_{x=1} = \left.\frac{\partial \log \mathcal{L}(c, p)}{\partial \log c}\right|_{c=c^\star(p)} \tag{38}$$

$$= -\delta. \tag{39}$$

Therefore, all curves collapse to first order around $x = 1$ as in Equation (27). □

**Remark.** A power-law Pareto frontier is not only necessary for full collapse but also already enforces a weaker, first–order form of collapse. Theorem E.1 assumes the compute–optimal point lies in the *interior* of each loss curve. This condition can fail for learning rate schedules that reach $\eta = 0$ after finitely many steps, because the optimum may then coincide with the boundary of the curve, where the envelope theorem tangency no longer applies. Such schedules are used throughout our experiments and are common in practice. Extension of the proof to handle these boundary-optimal schedules would be interesting.

# F. Collapse for General Sum-of-Power-Laws Loss Curves

**Theorem F.1.** *Suppose the loss curve is given by*

$$L(t, p) = L_0 + \sum_{i=1}^{m} a_i t^{-\mu_i} p^{-\nu_i}, \quad a_i > 0, \ \mu_i, \nu_i \geq 0, \tag{40}$$

*with at least one of $\mu_i, \nu_i$ positive for every $i$ (else absorb the term into $L_0$). Let $t^\star(p) = \kappa p^\gamma$ with $\kappa > 0, \gamma > 0$ be the asymptotic compute-optimal training horizon, and define the total exponent $\beta_i := \mu_i \gamma + \nu_i$ and $b_i := a_i \kappa^{-\mu_i}$. Without loss of generality, assume $\beta_i$'s are sorted in non-decreasing order. Then,*

*1. Compute-optimality forces a tie. At least two $\beta_i$'s share the minimum:*

$$\beta_1 = \beta_2 = \cdots = \beta_k < \beta_{k+1} \leq \cdots \leq \beta_m, \quad k \geq 2. \tag{41}$$

*2. Asymptotic collapse. The normalized loss curve*

$$\ell(x, p) := \frac{L(xt^\star(p), p) - L_0}{L(t^\star(p), p) - L_0}. \tag{42}$$

*is given by*

$$\ell(x, p) = \frac{\sum_{i=1}^{k} b_i x^{-\mu_i}}{\sum_{i=1}^{k} b_i} + O(p^{-\epsilon}), \quad \epsilon := \beta_{k+1} - \beta_1 > 0, \tag{43}$$

*independent of $p$ up to finite-size error that decays as $O(p^{-\epsilon})$. If $k = m$, $\epsilon$ is taken to be $\infty$ (perfect finite-size collapse).*

*3. Locally fastest decay of finite-size error. Locally, $\gamma$ is the data exponent that achieves the fastest decay of the finite-size error as measured by $\epsilon$. In particular, $\epsilon = O(|\delta|)$ for any other data exponent $\gamma' = \gamma + \delta$ with $\delta \neq 0$, leading to more slowly decaying finite-size error and therefore a worse collapse.*

*4. Compute-optimality up to a constant suffices. Any training horizon that is a constant multiple of $t^\star(p)$ preserves the collapse, only changing the constants $b_i$ in Equation (43).*

*Proof.* **Compute-optimality forces a tie.** Fix the compute budget $c := 6tp$ and note $t(p) = c/(6p)$ so that $\frac{dt}{dp} = -t/p$. With $\beta_i := \mu_i\gamma + \nu_i$ and $b_i := a_i\kappa^{-\mu_i}$,

$$\frac{dL}{dp} = \sum_{i=1}^{m} a_i\left(\frac{\partial}{\partial p} + \frac{dt}{dp}\frac{\partial}{\partial t}\right)t^{-\mu_i}p^{-\nu_i} \tag{44}$$

$$= \sum_{i=1}^{m} a_i\left(-\frac{\nu_i}{p} + \frac{t}{p}\frac{\mu_i}{t}\right)t^{-\mu_i}p^{-\nu_i} \tag{45}$$

$$= \frac{1}{p}\sum_{i=1}^{m} a_i(\mu_i - \nu_i)t^{-\mu_i}p^{-\nu_i} \tag{46}$$

$$= \frac{1}{p}\sum_{i=1}^{m} b_i(\mu_i - \nu_i)p^{-\beta_i}. \tag{47}$$

If $\beta_1 < \beta_2$, the leading term $b_1(\mu_1 - \nu_1)p^{-\beta_1}$ cannot cancel the rest for asymtotically large $p$, contradicting $\frac{dL}{dp} = 0$ required by compute-optimality. Hence at least two indices share the minimum exponent, yielding Equation (41).[1]

**Asymptotic collapse.** We compute $\ell(x, p)$ explicitly. First, evaluate the loss at the optimal horizon:

$$L(t^\star(p), p) - L_0 = \sum_{i=1}^{m} a_i(t^\star(p))^{-\mu_i}p^{-\nu_i} = \sum_{i=1}^{m} a_i(\kappa p^\gamma)^{-\mu_i}p^{-\nu_i} \tag{48}$$

$$= \sum_{i=1}^{m} a_i\kappa^{-\mu_i}p^{-\mu_i\gamma-\nu_i} = \sum_{i=1}^{m} b_ip^{-\beta_i}. \tag{49}$$

Since $\beta_1 = \beta_2 = \cdots = \beta_k < \beta_{k+1} \leq \cdots \leq \beta_m$, we can factor out $p^{-\beta_1}$:

$$L(t^\star(p), p) - L_0 = p^{-\beta_1}\left(\sum_{i=1}^{k} b_i + \sum_{i=k+1}^{m} b_ip^{-(\beta_i-\beta_1)}\right) \tag{50}$$

$$= p^{-\beta_1}\left(\sum_{i=1}^{k} b_i\right)\left(1 + O\left(p^{-(\beta_{k+1}-\beta_1)}\right)\right). \tag{51}$$

Similarly, for $t = xt^\star(p)$:

$$L(xt^\star(p), p) - L_0 = \sum_{i=1}^{m} a_i(xt^\star(p))^{-\mu_i}p^{-\nu_i} = \sum_{i=1}^{m} a_ix^{-\mu_i}(t^\star(p))^{-\mu_i}p^{-\nu_i} \tag{52}$$

$$= \sum_{i=1}^{m} b_ix^{-\mu_i}p^{-\beta_i} = p^{-\beta_1}\left(\sum_{i=1}^{k} b_ix^{-\mu_i}\right)\left(1 + O\left(p^{-(\beta_{k+1}-\beta_1)}\right)\right). \tag{53}$$

Taking the ratio gives:

$$\ell(x, p) = \frac{p^{-\beta_1}\left(\sum_{i=1}^{k} b_ix^{-\mu_i}\right)\left(1 + O\left(p^{-(\beta_{k+1}-\beta_1)}\right)\right)}{p^{-\beta_1}\left(\sum_{i=1}^{k} b_i\right)\left(1 + O\left(p^{-(\beta_{k+1}-\beta_1)}\right)\right)} = \frac{\sum_{i=1}^{k} b_ix^{-\mu_i}}{\sum_{i=1}^{k} b_i} + O\left(p^{-(\beta_{k+1}-\beta_1)}\right). \tag{54}$$

This produces Equation (43) with $\epsilon = \beta_{k+1} - \beta_1 > 0$.

**Locally fastest decay of finite-size error.** Let $\gamma$ be the optimal data exponent and perturb it by $\delta$, writing $\gamma' = \gamma + \delta$. For a small enough $|\delta| > 0$, the previously tied lowest exponents $\beta_1, \ldots, \beta_k$ split into distinct values $\beta_1', \ldots, \beta_k'$, which remain

---

[1] Here we assumed not all $\mu_i - \nu_i \neq 0$ for $i = 1, \ldots, k$, else these terms would not affect $dL/dp$. If this is not true, then the loss $L(t, p)$ is not interesting because the it would asymptotically be a function of compute $c = 6tp$ alone, independent of how we allocate $c$ between $t$ and $p$.

the lowest $k$ exponents (since $\delta$ is small), and the gap between the lowest and second-lowest grows as $O(|\delta|)$, which is strictly smaller than the previous gap $\epsilon = \beta_{k+1} - \beta_1 > 0$ for sufficiently small $\delta$. Therefore, locally $\gamma$ maximizes the decay exponent of the finite-size error, i.e., it gives the best collapse locally.

**Compute-optimality up to a constant suffices.** Replacing $t^\star$ by $\lambda t^\star$ multiplies each $b_i$ by $\lambda^{-\mu_i}$ and leaves the rest of the proof unchanged. $\qquad\square$

**Remarks.**

1. Compute-optimal data exponent implies asymptotic collapse, but the converse is not necessarily true when $m \geq 3$, since there can be multiple choices of $\gamma$ that lead to balanced dominant power laws, which imply collapse, but only one of them can be compute-optimal.

2. In general, when $m > 2$, asymptotic instead of exact collapse is the best we can hope for. But an asymptotic collapse alone is not that interesting, since under any choice of the data exponent $\gamma$ only the terms with the lowest $\beta_i$ enter the asymptotic normalized loss curve. For example, if $L(t, p) = t^{-\mu} + p^{-\nu}$, any $\gamma > \nu/\mu$ will cause only the $p^{-\nu}$ term to dominate, leading to $\ell(x, p) \to 1$, whereas any $\gamma < \nu/\mu$ will cause $t^{-\mu}$ to dominate, leading to $\ell(x, p) \to x^{-\mu}$. The latter case is similar to the infinite-width limit in neural networks, where under $t = \Theta(1)$, the loss curves become bottlenecked by training time alone and not model size (Vyas et al., 2023; Bordelon & Pehlevan, 2022; Yang & Littwin, 2023). What is interesting about the collapse that happens under compute-optimal training is that $\gamma$ is tuned (to $\nu/\mu$ in this example) so that more than one such term exists, so the collapse reflects a balanced scaling of both training time and model size. This fine balance is also why minor perturbations to $\gamma$ from the optimal value can significantly disrupt the collapse, which is not true if there is only one dominant power law.

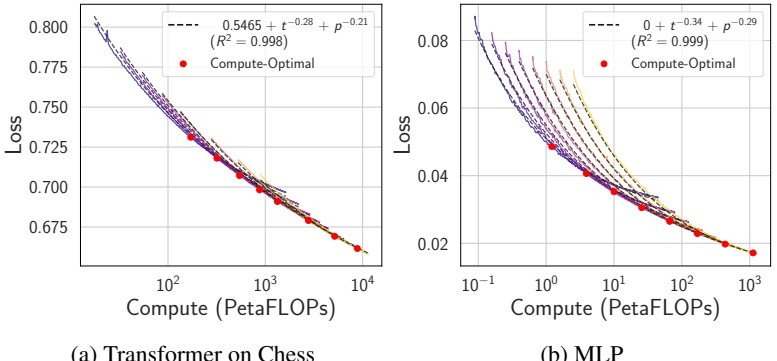

(a) Transformer on Chess        (b) MLP

Figure 11: **Sum-of-power-laws fit on additional datasets.** Both tasks have loss curves that can be approximated by the sum of two power laws when using a constant learning rate schedule. Fitted constant multipliers are not shown in the legend. To not fit to early-time transients, we fit all steps after the first 0.1B tokens for MLPs, and all steps after the first $0.1\times$ the compute-optimal horizon for chess. The fit for chess is worse than the other datasets.

# G. A Perturbative Model of Learning Rate Schedules

Let $w'$ denote the parameter trajectory under the influence of gradient noise. The dynamics of stochastic gradient descent in gradient flow time are given by

$$\frac{dw'}{d\tau} = -\Big(\nabla L(w') + \Sigma^{1/2}(w')\xi(\tau)\Big), \tag{55}$$

with noise correlation $\mathbb{E}[\xi(\tau)\xi(\tau')^\top] = \eta(\tau)\delta(\tau - \tau')$. For convenience, we rewrite this using $\xi(\tau) = \eta^{1/2}(\tau)\tilde{\xi}(\tau)$ so that

$$\frac{dw'}{d\tau} = -\Big(\nabla L(w') + \eta^{1/2}(\tau)\Sigma^{1/2}(w')\tilde{\xi}(\tau)\Big), \tag{56}$$

where now $\mathbb{E}[\tilde{\xi}(\tau)\tilde{\xi}(\tau')^\top] = \delta(\tau - \tau')$.

Our strategy is to solve $w'(\tau)$ as $w(\tau) + \delta w(\tau)$ where $w(\tau)$ is the deterministic trajectory satisfying $\frac{dw}{d\tau} = -\nabla L(w)$, up to leading order in the gradient noise scale $\eta \Sigma$.

Letting $\delta w := w' - w$, $g := \nabla L$ and taking the difference of the two differential equations:

$$\frac{d(\delta w)}{d\tau} = -\Big(g(w') - g(w) + \eta^{1/2}(\tau)\Sigma^{1/2}(w')\tilde{\xi}(\tau)\Big). \tag{57}$$

At first order,

$$g(w') \approx g(w) + H(w)\delta w \tag{58}$$

where $H(w) = \nabla^2 L(w)$ is the Hessian.

Our SDE for $\delta w$ becomes:

$$\frac{d(\delta w)}{d\tau} = -H(w)\delta w - (\eta\Sigma)^{1/2}\tilde{\xi}(\tau) \tag{59}$$

We define the propagator $G(\tau, s)$ that satisfies:

$$\frac{dG(\tau, s)}{d\tau} = -H(w(\tau))G(\tau, s) \tag{60}$$

with $G(s, s) = I$.

For time-dependent $H(w(\tau))$, the propagator is:

$$G(\tau, s) = \mathcal{T}\exp\left(-\int_s^\tau d\lambda\, H(w(\lambda))\right) \tag{61}$$

where $\mathcal{T}$ denotes time-ordering.

Assuming the initial perturbation $\delta w(0) = 0$, the solution for $\delta w$ is:

$$\delta w(\tau) = -\int_0^\tau ds\, G(\tau, s)(\eta\Sigma)^{1/2}(s)\tilde{\xi}(s). \tag{62}$$

Now expanding $L(w') = L(w + \delta w)$ to second order in $\delta w$ gives

$$\begin{aligned}
\delta L(\tau) &= L(w'(\tau)) - L(w(\tau)) \\
&\approx g(w(\tau))^\top \delta w(\tau) + \frac{1}{2}\delta w(\tau)^\top H(w(\tau))\, \delta w(\tau).
\end{aligned} \tag{63}$$

Since $\mathbb{E}[\tilde{\xi}(s)] = 0$ and $\delta w$ is linear in $\tilde{\xi}$, $\mathbb{E}[\delta w(\tau)] = 0$, so

$$\mathbb{E}\big[g(w(\tau))^\top \delta w(\tau)\big] = 0. \tag{64}$$

Thus the leading non-vanishing contribution to the expected loss shift comes from the quadratic term.

Using the solution for $\delta w$,

$$\delta w(\tau)\,\delta w(\tau)^\top = \int_0^\tau ds \int_0^\tau du\, G(\tau, s)(\eta\Sigma)^{1/2}(s)\tilde{\xi}(s)\tilde{\xi}(u)^\top(\eta\Sigma)^{1/2}(u)G(\tau, u)^\top. \tag{65}$$

Taking the expectation with $\mathbb{E}[\tilde{\xi}(s)\tilde{\xi}(u)^\top] = \delta(s - u)\, I$ gives

$$\mathbb{E}\big[\delta w(\tau)\,\delta w(\tau)^\top\big] = \mathbb{E}\left[\int_0^\tau ds\, G(\tau, s)\,\eta(s)\Sigma(w'(s))\,G(\tau, s)^\top\right]. \tag{66}$$

Substituting this into the quadratic term gives

$$\mathbb{E}[\delta L(\tau)] = \frac{1}{2}\mathbb{E}\left[\int_0^\tau ds \ \mathrm{Tr}\left[H(w(\tau))\, G(\tau, s)\, \eta(s)\Sigma(w'(s))\, G(\tau, s)^\top\right]\right]. \tag{67}$$

Using $\mathrm{Tr}[ABC] = \mathrm{Tr}[CAB]$,

$$\mathbb{E}[\delta L(\tau)] = \frac{1}{2}\mathbb{E}\left[\int_0^\tau ds \ \mathrm{Tr}\left[G(\tau, s)^\top H(w(\tau))\, G(\tau, s)\eta(s)\Sigma(w'(s))\right]\right]. \tag{68}$$

This equation is the exact leading-order expression for the noise-induced change in expected loss, which is the equivalent of $\mathcal{E}(\tau)$ in Equation (10). The deterministic loss $L(w(\tau))$ now plays the role of $\mathcal{F}(\tau)$ in Equation (14). Conceptually, the derivation shows that—although the full dynamics are non-linear and the loss is not assumed quadratic—the perturbation generated by small gradient noises behaves in a simple, linear-quadratic fashion: the weight perturbation $\delta w$ is linear in the injected noise, and the resulting loss shift $\delta L$ is quadratic in that perturbation.

Consequently the derivation and final formula completely mirrors the familiar quadratic-loss result, the only difference being that the constant Hessian $H$ is now replaced by the time-dependent Hessian $H(w(\tau))$ carried along the deterministic trajectory. In other words, small gradient noise "sees" the network through an instantaneous linearization, so all schedule effects enter through the propagator $G(\tau, s)$, the local Hessian, and the noise covariance, exactly as in the linear case.

**Relation to Stochastic Asymptotic Expansion.** Li et al. (2017) used stochastic asymptotic expansion to expand the dynamics in orders of $\eta^{1/2}$, but they treat the gradient-noise covariance differently. The stochastic asymptotic expansion in Li et al. (2017) first expands the diffusion term along the deterministic path, so $\Sigma$ can be propagated analytically order by order. Our derivation instead keeps the exact $\Sigma(w'(s))$ inside the leading-order integral, allowing an empirically measured covariance to be inserted without further approximation, at the cost of analytic closure.

**Slow-Variation and Late-Time Limit.** As in the quadratic loss case, we can simplify the result under an adiabatic approximation where the Hessian, schedule, and noise covariance changes slowly compared to the time-scale set by the instantaneous Hessian. Specifically, if $H(w(t)) \approx H(w(\tau)) \coloneqq H$ over the support of $G(\tau, s)^\top H(w(\tau))\, G(\tau, s)$, then $G(\tau, s) \approx e^{-H(\tau-s)}$ and $G(\tau, s)^\top H G(\tau, s) \approx H\, e^{-2H(\tau-s)}$. Assuming the exponential decay is fast compared to the variation of the noise scale $\eta\Sigma$, and taking $\tau \to \infty$, we have

$$\mathbb{E}[\delta L(\tau)] \approx \frac{1}{4}\,\mathrm{Tr}\left[\eta(\tau)\bar{\Sigma}(w'(\tau))\right], \tag{69}$$

which agrees with the expression for $\mathcal{E}(\tau)$ in Equation (14) for the quadratic loss setting.

**Adaptive Optimizers.** When using adaptive optimizers with a preconditioner $P(t)$, the SDE becomes

$$\frac{dw'}{dt} = -\eta(t)P^{-1}(t)\Big(\nabla L(w') + \Sigma^{1/2}(w')\xi(t)\Big). \tag{70}$$

The gradient flow time SDE in Equation (56) becomes

$$\frac{dw'}{d\tau} = -P^{-1}(\tau)\Big(\nabla L(w') + \eta^{1/2}(\tau)\Sigma'^{1/2}\tilde{\xi}(\tau)\Big), \tag{71}$$

where we abbreviated $\Sigma(w')$ as $\Sigma'$. If the preconditioner varies slowly, the dynamics can be treated as if there is no preconditioner, but in a transformed coordinate system:

$$\tilde{w}'(t) = P^{1/2}(t)\, w'(t). \tag{72}$$

Differentiating and neglecting the $O(\dot{P})$ term gives

$$\frac{d\tilde{w}'}{d\tau} = P^{1/2}\frac{dw'}{d\tau} = -P^{-1/2}\Big(\nabla L(w') + \eta^{1/2}\Sigma'^{1/2}\tilde{\xi}\Big) \tag{73}$$

$$= -\tilde{g}(\tilde{w}') - \underbrace{(\eta P^{-1}\Sigma')^{1/2}\tilde{\xi}}_{\text{noise}}. \tag{74}$$

The deterministic trajectory in this coordinate system ($\tilde{w}(t) = P^{1/2}(t)\, w(t)$) satisfies

$$\frac{d\tilde{w}}{d\tau} = -\tilde{g}(\tilde{w}). \tag{75}$$

To dervie the SDE for $\delta\tilde{w} := \tilde{w}' - w$, at first order, we have

$$\tilde{g}(\tilde{w}') - \tilde{g}(\tilde{w}) = P^{-1/2}H\delta w = P^{-1/2}HP^{-1/2}\delta\tilde{w} := \tilde{H}\delta\tilde{w}, \tag{76}$$

where $\tilde{H}$ is the preconditioned Hessian. Therefore,

$$\frac{d\delta\tilde{w}}{d\tau} = -\tilde{H}\delta\tilde{w} - (\eta P^{-1}\Sigma')^{1/2}\tilde{\xi}, \tag{77}$$

It is also easy to show that the preconditioned Hessian indeed governs the leading order perturbation to the expected loss in the transformed coordinates

$$\mathbb{E}[\delta L] \approx \frac{1}{2}\mathbb{E}[\delta\tilde{w}^\top \tilde{H}\delta\tilde{w}]. \tag{78}$$

Given Equation (77) and Equation (78), all steps in the previous derivation now applies after swapping $H \to \tilde{H}$ and $\Sigma' \to P^{-1/2}\Sigma'P^{-1/2} := \tilde{\Sigma}'$. The final result for the noise-induced loss perturbation in the slow-variation and late-time limit is

$$\mathbb{E}[\delta L(\tau)] \approx \frac{1}{4}\operatorname{Tr}\left[\eta(\tau)\bar{\tilde{\Sigma}}(w'(\tau))\right]. \tag{79}$$

**Limitations.** It is worth highlighting some limitations of our analysis. First, due to the non-linear nature of neural networks, it is known that gradient flow can fail to model full-batch gradient descent with a finite step size, which exhibits effects such as the Edge of Stability (Cohen et al., 2021; 2022; 2024). Similarly, in the stochastic case, there is a strong coupling between learning rate, batch size, and the Hessian spectrum (Agarwala & Pennington, 2024). These phenomena suggest that it is unlikely that we can fully model the effect of a time-varying learning rate schedule simply as injecting a schedule-dependent noise component on top of the deterministic trajectory that is itself independent of the schedule, as changing the learning rate also pushes the model into different regions of the parameter space based on curvature (though it is possible that the leading-order perturbation theory can already capture this effect to some extent).

Second, when dealing with adaptive optimizers, we assumed the preconditioner stays the same (as a function of $\tau$) between the deterministic and stochastic trajectories. For typical optimizers, such as Adam (Diederik P. Kingma, 2015), this assumption is not correct as $P$ depends on the gradient covariance, which can differ between the two trajectories, i.e., $\Sigma(w(\tau)) \neq \Sigma(w'(\tau))$. This can introduce an additional term in the SDE for $\delta\tilde{w}$, present even at first order, which we did not model.

The empirical effectiveness of our model for predicting the loss curves under different schedules suggests it is nevertheless on the right track, and that there may be other ways to derive similar or improved predictions with more accurate assumptions.

# H. Computing $\tilde{\Delta}$ to Leading Order

Let $\psi(\tau) = \frac{\mathcal{L}(\tau) - \bar{\mathcal{L}}(\tau)}{\bar{\mathcal{L}}(\tau)}$ and define

$$\tilde{\Delta}^2(\tau) = \mathbb{E}\left[(\psi(\tau) - \psi(\tau^\star))^2\right] \tag{80}$$

with $\tau^\star = \tau + \delta\tau$ and $0 < \delta\tau \ll 1$. Because $\bar{\mathcal{L}}(\tau^\star)^{-1} = \bar{\mathcal{L}}(\tau)^{-1} + O(\delta\tau)$,

$$\tilde{\Delta}^2(\tau) = \bar{\mathcal{L}}^{-2}(\tau)\,\mathbb{E}\left[(\Delta\mathcal{L}(\tau) - \Delta\mathcal{L}(\tau^\star))^2\right] + O(\delta\tau^2), \tag{81}$$

where $\Delta\mathcal{L}(\tau) = \mathcal{L}(\tau) - \bar{\mathcal{L}}(\tau)$. For the quadratic model,

$$\Delta\mathcal{L}(\tau) = g(\tau)^\top \Delta w(\tau) \tag{82}$$

to first order in

$$\Delta w(\tau) = \int_0^\tau ds\, e^{-H(\tau-s)} \Sigma^{1/2}(s)\, \xi(s) \tag{83}$$

with

$$\mathbb{E}[\xi(s)\xi(s')] = \eta(s)\delta(s-s')I. \tag{84}$$

Splitting the upper limit at $\tau$ and expanding $g(\tau^\star) = g(\tau) + O(\delta\tau)$ gives

$$\Delta\mathcal{L}(\tau) - \Delta\mathcal{L}(\tau^\star) = \underbrace{g(\tau)^\top \int_\tau^{\tau^\star} ds\, e^{-H(\tau^\star-s)} \Sigma^{1/2}(s)\, \xi(s)}_{O(\delta\tau^{1/2})} + O(\delta\tau) \tag{85}$$

where the remainder collects two subleading $O(\delta\tau)$ contributions:

$$R_1 := g(\tau)^\top \int_0^\tau ds\, \left[ e^{-H(\tau^\star-s)} - e^{-H(\tau-s)} \right] \Sigma^{1/2}(s)\, \xi(s), \tag{86}$$

$$R_2 := [g(\tau^\star) - g(\tau)]^\top \Delta w(\tau) = \dot{g}(\tau)^\top \Delta w(\tau)\, \delta\tau + O(\delta\tau^2). \tag{87}$$

Using $e^{-H(\tau^\star-s)} = I + O(\delta\tau)$, $\eta(s) = \eta(\tau) + O(\delta\tau)$, and $\bar{\Sigma}(s) = \bar{\Sigma}(\tau) + O(\delta\tau)$ inside the leading-order integral,

$$\mathbb{E}\left[ (\Delta\mathcal{L}(\tau) - \Delta\mathcal{L}(\tau^\star))^2 \right] = g(\tau)^\top \int_\tau^{\tau^\star} ds\, \eta(s)\, e^{-2H(\tau^\star-s)} \bar{\Sigma}(s) g(\tau) + O(\delta\tau^2) \tag{88}$$

$$= g(\tau)^\top \eta(\tau)\bar{\Sigma}(\tau) g(\tau)\, \delta\tau + O(\delta\tau^2). \tag{89}$$

Substituting this into the expression for $\tilde{\Delta}^2(\tau)$ yields the desired result

$$\tilde{\Delta}^2(\tau) = \bar{\mathcal{L}}^{-2}(\tau)\, g(\tau)^\top \eta(\tau)\bar{\Sigma}(\tau) g(\tau)\, \delta\tau + O(\delta\tau^2). \tag{90}$$

## I. Additional Results on Learning Rate Schedules

**MLP fits.** Figure 12 shows our predictions for MLP loss curves. With a single $\alpha = 0.26$ (very close to $1/4$), we obtain excellent fits across schedules, model sizes, and training horizons.

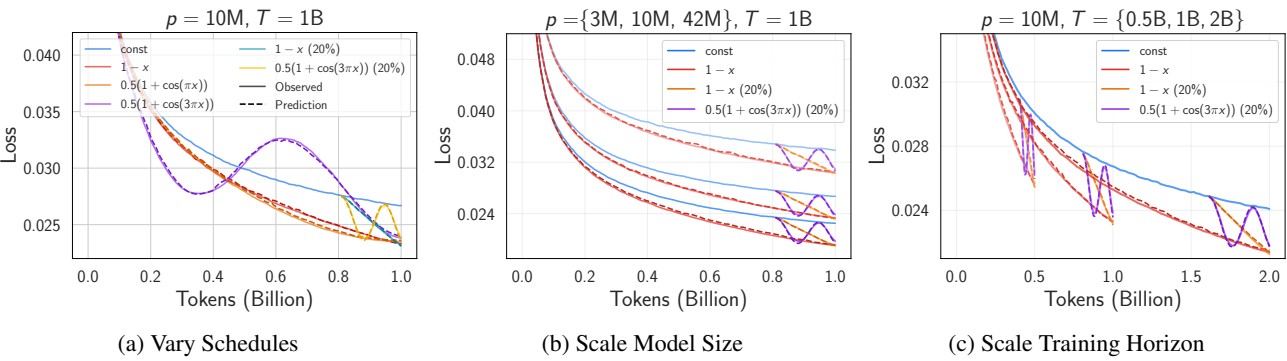

(a) Vary Schedules     (b) Scale Model Size     (c) Scale Training Horizon

Figure 12: **A simple model predicts MLP loss curves trained across learning rate schedules, model sizes $p$, and training horizons $T$ on the syntheticic regression task.** Dashed curves show the predicted loss as $L'(\tau) = L(\tau) + \alpha\, \delta\eta(\tau)\, \mathrm{Tr}(\Sigma'(\tau))$ (Equation (18)). $\alpha$ is the only free parameter and is set to 0.26. Each curve is smoothed with an exponential moving average with half life equal to 1% of total steps.

**Effect of Dropping** $\eta\, \mathrm{Tr}(\delta\Sigma)$. Figure 13 (top row) shows $\delta\eta\, \mathrm{Tr}(\Sigma')$ is typically 3 to 10 times than $\eta\, \mathrm{Tr}(\delta\Sigma)$ in absolute value in both CIFAR-5M transformer and MLP regression experiments. Since we decay the learning rate to zero ($\delta\eta$ is

comparable $\eta$), this means the gradient covariance does not change much between the constant learning rate and other schedules, which can happen if the Hessian trace does not change much. The fact that this ratio is only moderately large shows that the gradient covariance or the Hessian trace did change considerably due to decaying the learning rate, which is what we expect due to a generically inverse relation between learning rate and Hessian eigenvalues (Cohen et al., 2021; Agarwala & Pennington, 2024). However, somewhat puzzlingly, we found that including the smaller term $\eta \operatorname{Tr}(\delta\Sigma)$ can produce worse fits, particularly for the slow oscillatory schedule ($0.5 \cos(3\pi x)$), and makes the optimal constant $\alpha$ more schedule-dependent.

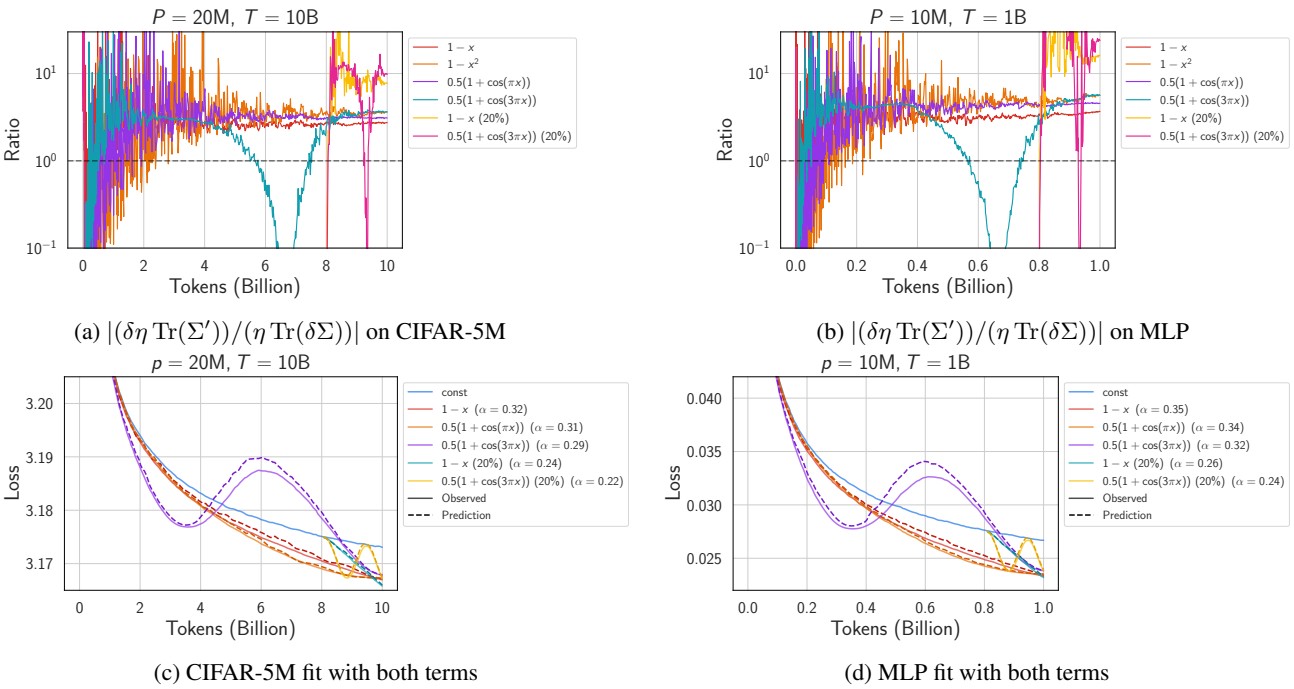

(a) $|(\delta\eta \operatorname{Tr}(\Sigma'))/(\eta \operatorname{Tr}(\delta\Sigma))|$ on CIFAR-5M

(b) $|(\delta\eta \operatorname{Tr}(\Sigma'))/(\eta \operatorname{Tr}(\delta\Sigma))|$ on MLP

(c) CIFAR-5M fit with both terms

(d) MLP fit with both terms

Figure 13: Out of the two terms that make up $\operatorname{Tr}(\delta(\eta\Sigma))$, the term $\eta \operatorname{Tr}(\delta\Sigma)$ is typically 3 to 10 times smaller than $\delta\eta \operatorname{Tr}(\Sigma')$ (top row). Moreover, including it sometimes produces a worse fit and make the optimal $\alpha$ vary more across schedules (bottom row). We determine $\alpha$ for each schedule by matching the prediction with the observation at the end point.

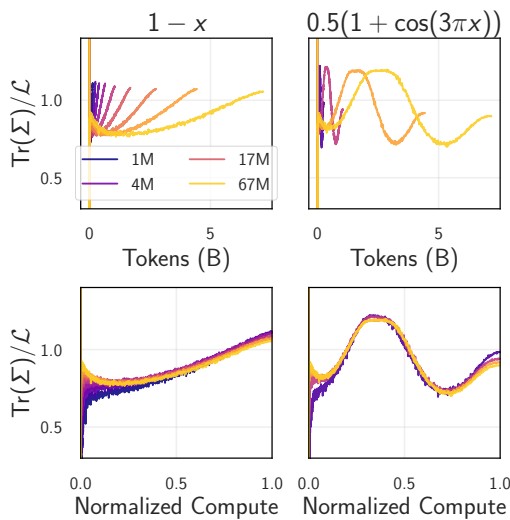

Figure 14: **Universality of gradient noise in MLPs.** Fixing a learning rate schedule, the ratio $\operatorname{Tr}(\Sigma)/\mathcal{L}$ is approximately a function of normalized compute alone, independent of model size. On this regression task, the estimated irreducible loss is negligible so $\mathcal{L} \approx L$.

