# OpenReview forum: "Scaling Collapse Reveals Universal Dynamics in Compute-Optimally Trained Neural Networks"
_ICML.cc/2025/Conference — ICML 2025 oral_

### Official Review · Reviewer_oTD2 · 2025-03-07

**Overall Recommendation:** 4

**Summary:**

Assuming that any model is trained for a number of iterations that are compute-optimal given its size, this paper shows, both empirically and theoretically, that the training curves of models of different widths are identical up to an affine transformation.
Deviations due to randomness in the training procedure are larger than those across model width.
Furthermore the paper shows, both empirically and theoretically, that the result holds across a variety of learning rate schedules.
These results can be used to improve scaling predictions.

**Claims And Evidence:**

This work can potentially help scaling up models more accurately.
The combination of empirical and theoretical results, and the agreement between the two, is compelling.
The density and significance of results is high.

**Essential References Not Discussed:**

OK

**Experimental Designs Or Analyses:**

OK

**Methods And Evaluation Criteria:**

In CIFAR-5M experiments, the model width varies very little, less than a factor of three, which corresponds to a one order magnitude variation of the parameter count. Fitting power laws with such small variation is unreliable.

**Other Comments Or Suggestions:**

NA

**Other Strengths And Weaknesses:**

NA

**Questions For Authors:**

If it’s true that the primary effect of learning rate decay is annealing SGD noise, then can we determine what is the best way of decaying the learning rate?

**Relation To Broader Scientific Literature:**

OK

**Theoretical Claims:**

It remains unclear how much useful is Eq.6. The proof of collapse in Appendix G assumes that the sum in Eq.6 can be approximated by two terms. Eq.7 is also a special case of Eq.6 with two terms of the sum only. So it remains unclear whether we need Eq.6 at all.

---

> ### Author Rebuttal · Authors · 2025-04-01
>
> Thank you for your feedback and supportive review. We have attached some additional figures [here](https://drive.google.com/file/d/1ZkobNTqh90nnUcunKqUT2Dyx3T4zAb5O/view), and address your specific questions below.
>
> **On the limited range of widths in CIFAR-5M experiments.**
> We acknowledge the limitations of this experiment, which was a combination of our computational constraints as well as the small size of the dataset itself. We set the maximum width to 2048 in the CIFAR-5M experiments to 1) limit the computational cost of our experiments, and 2) avoid excessive repetition of the training data. Based on the estimated compute-optimal training horizon, the width 2048 model already required training on the CIFAR-5M dataset (which only has 5M training images) for over 10 epochs. A larger dataset is necessary to train even wider models to compute-optimality without overfitting effects.
>
> We note that our MLP experiment does not suffer from this data constraint, where we scaled up the width by 8x from 512 to 4096, corresponding to a 64x scale up in model size, and demonstrated supercollapse across all models.
>
> In addition, we demonstrate in Figure 1 (middle, right) [here](https://drive.google.com/file/d/1ZkobNTqh90nnUcunKqUT2Dyx3T4zAb5O/view) that supercollapse occurs in two additional data modalities: chess game string representations and chemical structures, and along one more scaling dimension: depth, further establishing the generality of our observations.
>
> **On the utility of Equation 6.**
> We emphasize that our claim regarding the dominance of two terms in Equation 54 is mathematically valid in the asymptotic limit of large $D$, not merely a convenient approximation. This generalized analysis of power law loss curves extends the connection between supercollapse and compute-optimal scaling beyond the simplified form in Equation 7. This extension is particularly important because recent theoretical work [1] demonstrates that this broader class of loss curves naturally emerges in high-dimensional regression models, whereas the two-term power law in Equation 7 fails to capture significant regions of the phase diagram (Figure 2, [1]).
>
> To empirically validate our theoretical findings, Figure 6 attached via [this link](https://drive.google.com/file/d/1ZkobNTqh90nnUcunKqUT2Dyx3T4zAb5O/view) presents additional experiments using the Power Law Random Features (PLRF) model introduced in Section 3.1. Unlike our setup in Figure 4, we specifically selected values of $\alpha$ and $\beta$ to produce loss curves asymptotically described by three power laws (detailed in [1]), which cannot be accurately represented by Equation 6. These results confirm that supercollapse occurs exactly as predicted by our analysis in Appendix G. Without our generalized framework, we would lack the theoretical foundation to explain supercollapse in these more complex settings.
>
> **On determining the optimal decay schedule**
> For a fixed $\hat{K}$, equation 10 suggests that decaying the learning rate as late as possible to maximize total gradient flow time is best. However, as we show in Figure 5 and Figure 9 in the paper, the best fit $\hat{K}$ has non-trivial dependence on the schedule and correlates with the preconditioned NTK trace, which tends to be higher for schedules that start decaying earlier. This gives a tradeoff between progress in gradient flow time (favoring later decay) and more favorable geometry (favoring early decay). Understanding this tradeoff requires a deeper understanding of $\hat{K}$, its dynamics, and its relationship to the schedule. We believe this is an exciting direction for future research on optimizing learning rate schedules.
>
> Thank you again for your review. We hope we have addressed your questions. Please let us know if we can clarify anything further.
>
> [1] Paquette et al. "4+3 phases of compute-optimal neural scaling laws."

---

### Official Review · Reviewer_NFpH · 2025-03-13

**Overall Recommendation:** 4

**Summary:**

This paper introduces the concept of "supercollapse," where the loss curves of networks trained under compute-optimal conditions collapse to a universal curve after affine rescaling. The authors demonstrate this phenomenon across various architectures (transformers and mlps, with different dim sizes) and learning rate schedules, providing both empirical evidence and theoretical insights into why and how this collapse occurs. This study contributes to the understanding of scaling behaviors in networks and suggests new ways to optimize training processes.

**Claims And Evidence:**

Pros: The study provides detailed experimental setups and rigorous theoretical analysis to support its claims, including:
+ Supercollapse is observable across various architectures and learning rate schedules.
+ Accurate estimation of compute-optimal parameters is essential for observing supercollapse.

Cons: However, some concerns remain. The training datasets are somewhat limited, as only CIFAR-5M was used, with next-pixel predictions as the employed task. This does not fully align with the motivation to explore the dynamics of large models, particularly large language models (LLMs) or multimodal language models, which are the primary focus. Ideally, both inputs and outputs should consist of text tokens. It remains uncertain whether the conclusions drawn from CIFAR-5M and the next-pixel prediction task are applicable to LLM pre-training.

**Essential References Not Discussed:**

N/A

**Experimental Designs Or Analyses:**

The experimental design is solid and well-executed, both in terms of the model and the conditions. However, the inclusion of additional datasets, as previously mentioned, would further enhance the study.

**Methods And Evaluation Criteria:**

The methods are appropriate and clearly described, but the evaluation is weak in terms of the dataset.

**Other Comments Or Suggestions:**

N/A

**Other Strengths And Weaknesses:**

This paper attempts to address an important issue: predicting the universal curve of a model during training. It has the potential for a significant impact and could be highly beneficial for neural network training. However, due to the limited and insufficient diversity in the training data, the conclusions require further justification.

Additionally, there are a few typos and grammar errors:
+ Line 135: "80% of training" should be "80% of training."
+ Line 156: "denotes" should be "denote."

**Questions For Authors:**

N/A

**Relation To Broader Scientific Literature:**

This paper provides valuable insights that could support a wide range of model training-related tasks.

**Theoretical Claims:**

Theoretical analysis is provided to support the claims.

---

> ### Author Rebuttal · Authors · 2025-04-01
>
> Thank you for the thoughtful review.
>
> Your point about the limited diversity of the datasets is well taken. To address this point, we conducted additional experiments on two non-image domains: the [Lichess chess games dataset](https://huggingface.co/datasets/Lichess/chess960-chess-games) dataset of 22M chess games recorded in algebraic chess notation, and [SMILES-250M](https://huggingface.co/datasets/HoangHa/SMILES-250M), a large collection of chemical structure representations. In Figure 1 (middle, right) of the [linked PDF](https://drive.google.com/file/d/1ZkobNTqh90nnUcunKqUT2Dyx3T4zAb5O/view), we show the next-token prediction loss curves of compute-optimally trained transformers and their rescaled versions. Our results demonstrate that supercollapse occurs consistently across both these datasets. In addition, we showed that scaling depth also gives supercollapse in the MLP setting similar to scaling width (Figure 1 left). Combined with our original findings on CIFAR-5M, MLP regression, and high-dimensional linear models, these results provide compelling evidence that supercollapse is a general phenomenon spanning diverse data modalities and model architectures.
>
> Regarding LLM pre-training, computational constraints prevented us from conducting meaningful experiments. The smallest model in the Chinchilla scaling law [1] study has approximately 80M parameters, similar to our largest tested model. Our limited computation meant that we could not perform high-quality scaling studies in this setting. We believe that our consistent observation about supercollapse across multiple architectures (transformer, MLP, linear model) and data modalities (images, chess, chemical structures, regression), as well as our theoretical analysis establishing the connection between supercollapse and sum-of-power-law loss curves motivates future studies on LLM pre-training, which we hope to pursue in followup work.
>
> While we recognize the potential practical impact of our findings for LLM pre-training, we would like to emphasize that the primary contribution of this work is establishing supercollapse as a novel and intriguing phenomenon across diverse settings, and demonstrating that studying universality in loss curves offers a promising avenue for advancing our scientific understanding of scaling. We will revise the paper to articulate this scientific focus more clearly.
>
> Thank you again for your constructive feedback, which has helped improve the clarity and positioning of our paper. We will also correct the typos you mentioned in the revised version of the paper. We hope you will consider raising your score in light of our response and additional experiments.
>
> [1] Hoffmann et al. "Training compute-optimal large language models."

---

> > ### Comment · Reviewer_NFpH · 2025-04-04
> >
> > The new results are quite impressive. I would like to increase the score. Further testing across different modalities and datasets will undoubtedly enhance the overall quality of the paper.

---

### Official Review · Reviewer_hiQw · 2025-03-13

**Overall Recommendation:** 4

**Summary:**

The paper intvestigates the phenomenon of "supercollapse," where loss curves from compute-optimally trained neural networks collapse to a single universal curve, after an affine rescaling. This universality is observed across different model sizes and learning rate schedules, and it is characterized by deviations smaller than those caused by stochastic optimization noise. The authors prove that training to a Pareto frontier is necessary for supercollapse and show that learning rate schedules deform loss curves in a predictable, scale-invariant way. They validate their claims using transformer models on CIFAR-5M and MLPs on a synthetic regression task, showing that supercollapse occurs across architectures and tasks. The paper suggests that the study of full loss curves, instead than just the final loss, can improve scaling predictions and provide insights into the training dynamics of large models.

**Claims And Evidence:**

The claim are supported by extensive numerical evidence.

**Essential References Not Discussed:**

I am not aware of any essential references that have been omitted.

**Experimental Designs Or Analyses:**

I have no issues to discuss.

**Methods And Evaluation Criteria:**

The methods are appropriate for the problem.

**Other Comments Or Suggestions:**

- Typo on line 027: repetition of "the"
- Typo on line 135, second column: missing period
- Typo in eq. 2: there is a right square bracket without the corresponding left one
- Could you clarify the recursive definition of $k_i$ on line 138?

**Other Strengths And Weaknesses:**

Strengths: The paper introduces the concept of supercollapse and extends the literature on scaling laws by revealing universal dynamics in the full loss trajectories. It combines a theoretical analysis (showing the necessity of a power-law Pareto frontier) and empirical validation across architectures (transformers and MLPs) and tasks (CIFAR-5M and synthetic regression), as a support to its claims. The findings and insights provided by this work have practical implications for scaling predictions and optimization strategies, suggesting that small-scale experiments can predict the behavior of larger models.

Weaknesses: Some key quantities are not clearly defined in the main text (for instance $\gamma$ and $P_0$ on line 147, $k$ in equation 6, and $M$ on line 136). While some of them can be deduced from the context and others are clarified in the supplementary material, I would still suggest that the authors clarify all notation directly in the main text to ensure accessibility for the reader.

**Questions For Authors:**

I have no additional questions.

**Relation To Broader Scientific Literature:**

The key contributions of the paper relate to the literature on scaling laws and training dynamics in machine learning, expanding on previous work on empirically observed power-law relationships between compute, model size, and test risk. It goes further by uncovering a universal structure in full loss trajectories, moving beyond the usual focus on final loss values. The paper also links its empirical findings to theoretical research in high-dimensional optimization and random features.

**Theoretical Claims:**

I check the proof of Theorem 3.1 and I have no issues to discuss.

---

> ### Author Rebuttal · Authors · 2025-04-01
>
> Thank you for your careful reading of our draft and supportive review. We will update the paper to ensure that all notations are clearly defined in the main text and fix the typos you identified. Specifically regarding the definition of $k_i$, we first sample a scalar $s_i$ from the power-law distribution defined in L140 and then sample a random unit vector $v_i$ and define $k_i$ as $s \cdot v_i.$ This will be clarified in the text.
>
> We have attached a PDF [here](https://drive.google.com/file/d/1ZkobNTqh90nnUcunKqUT2Dyx3T4zAb5O/view) with additional experiments requested by other reviewers to further demonstrate the generality of supercollapse. If you are interested, the details can be found in our other rebuttals. Let us know if we can clarify anything further.

---

> > ### Comment · Reviewer_hiQw · 2025-04-09
> >
> > Thanks for your reply and additional experiments.

---

### Official Review · Reviewer_MwGt · 2025-03-14

**Overall Recommendation:** 5

**Summary:**

- When neural networks are trained under compute optimality, their loss curves across model widths collapses to a single universal curve under a simple affine rescaling
- The authors call this "supercollapse" because deviations between curves are smaller than fluctuations from multiple training runs (where the noise arises from random initialization and data ordering)
- Supercollapse occurs across various learning rate schedules (so long as they're decaying) and architectures, but requires accurate estimation of the compute-optimal data exponent and irreducible loss
- The authors prove that at power-law Pareto frontier is necessary (and in some cases, sufficient) for supercollapse to emerge
- Several other complementary results

## Update After Rebuttal

I think this is a great paper. I don't think it deserves to be an Oral (in my Claims and Evidence, I pinpointed two steps that I think this paper needed to do in order to be that outstanding) but I think the authors have done a wonderful job.

**Claims And Evidence:**

- Overall, I think this is a solid and thorough paper.
- In my review, I scrutinize some specific components. I welcome the authors to improve the paper based on my comments, or to convince me that my comments are mistaken. Given compelling evidence for either, I will increase my score.
- I think this paper could be an awarded paper if its practical relevance was increased. One way would be to show that supercollapse holds for language model ladders pretrained at scale, but this is likely an expensive request that the authors may be unable to afford. Another way would be to quantitatively compare how a “supercollapse-based” scaling law predictor compares to standard scaling law predictors, ideally in a backtested manner; the authors hint at this (“suggesting that supercollapse itself can be used to improve scaling predictions”) but stop short of actually applying this claim practically. I’m also unsure of whether a supercollapse predictor is feasible since Figure 3 suggests that $\gamma$ and $L_0$ are the most sensitive, and I would intuitively guess that these two parameters require large models to accurately estimate, but I could be wrong!
- To me, the point of highest uncertainty in this paper is its generality. My doubt stems from the fact the authors only swept fixed-depth networks. I couldn’t find a reason, I don’t understand why the authors did this, and there is now a question in my mind of whether supercollapse is as general as the authors claim it is.

**Essential References Not Discussed:**

None that spring to mind.

**Experimental Designs Or Analyses:**

- I can’t tell whether I’m reading too much into this, but the sentence “We scale the model size by increasing the width D and keep depth fixed.” raised three questions: (1) Why did the authors not scale the models with depth? (2) How do the results change (if at all) when depth is scaled with width or independently from width? (3) More generally, are there realistic scaling recipes under which supercollapse does not appear?
- Appendix C.1: I’m not quite sure I understand the methodology here. To confirm my understanding: (a) it appears that parameters were swept from ~7e6 to 7e7 for Transformer and from 2e6 to 7e7 for MLP? (b) For each parameter size, to vary the compute, presumably you swept a variety of tokens? What values of tokens were swept exactly, and why is the range of compute per parameter note the same? (c) Why were these models trained without learning rate decay? (d) In Figure 7, for each parameter value, I would expect to see some dot indicating the optimal compute for that parameter size, but I don’t; am I misunderstanding what this figure purports to show, or should the optimal compute at each parameter size be indicated by a unique marker? (e) I don’t know how much this matters, but how was the power law fit? Linear regression in log-log space, something more akin to Chinchilla with Huber loss, something else?
- Section 2.2: What is $\theta$? The fraction of optimal compute, yes? If so, it might be good to explicitly state this in the text.
- Section 2.2: nit: Right column, line 135: Missing period after “training”.
- Section 2.2: I feel like I’ve misunderstood something about the “last 80% of training”. Figure 1 (middle) suggests to me that the curves overlap for the last 90% of training. Am I misreading the data? Or where is this 80% figure coming from?
- Section 2.3: Line 174 left column “In Figure 1 (right)”, am I confused or should this refer to Figure 2?
- Figure 2: There appear to be no error bars or notions of uncertainty. Could the authors please repeat this experiment multiple times, or if that is too expensive, bootstrap it? I’m unsure whether the cross-D collapse deviation $\Delta(t)$ is indeed meaningfully smaller than the cross-seed fluctuation.
- Figure 2: What do the different hues correspond to?
- Line 193-195, left column: “This shows that the end point of a specific loss curve efficiently encodes much of the randomness throughout the trajectory.” I’m not quite sure I understand this sentence. Could the authors please clarify?
- Section 2.3: I'm uncertain whether the subsection's result is profound or trivial. We certainly expect this result at $\theta=1$. The authors point out, the interesting aspect is that it stays smaller over an $O(1)$ fraction of training time, but I imagine one could handwave that we expect smooth(ish) scaling trends.
- Figure 3 (c) Left: Could the authors please consider a larger range of $P_0$ values for the sensitivity analysis? I want to know how much $P_0$ can really be stretched before the collapse deviation increases substantially.
- (minor) Figure 9 (a): Why is variance so large in the vertical direction?

**Methods And Evaluation Criteria:**

Yes. I add specific comments in later sections.

**Edit: Based on the authors' response, I'm upgrading my score to a 5**

> $\theta$ refers to the fraction of optimal compute.

To clarify, I was suggesting that $\theta$ should be defined in the main text, likely in Section 2.2. It's possible I'm missing its definition in the main text, but I couldn't find one (and while $\theta$'s meaning can be inferred contextually, I think one should prefer being explicit).

**Other Comments Or Suggestions:**

- nit: “constan” on Line 533 should be “constant”
- nit: On line 437 left column, should "explaining why schedules at decay faster have..." be "explaining why schedules that decay faster have" ?

**Other Strengths And Weaknesses:**

See above.

**Questions For Authors:**

See earlier boxes.

**Relation To Broader Scientific Literature:**

I think this paper contextualizes itself well w.r.t. large scale empirical scaling laws e.g., Kaplan, Hoffman, and analytically tractable models, e.g., Paquette.

**Theoretical Claims:**

- Theorem 3.1 is a nice result.
- Section 3.1 Sum of Power Laws Collapse Exactly: To clarify my understanding, Equations 7-9 (inclusive) hold only for the PLRF model? If so, could the authors please help me understand why a power law compute-optimal Pareto frontier is only sufficient for collapse near $\theta=1$?
- Section 3.2 Line 321 Right Column: The manuscript states, “This result suggests that the primary effect of learning rate decay is to accelerate the loss curves by annealing the noise introduced by SGD.” I’m confused by this claim. I would think that the noise introduced by SGD is inherent to the dataset, the batch size and the model. How is the learning rate able to suppress noise in SGD?
- Equation 2: Since $s$ appears in the expectation and variance, I would expect it to appear inside the squared brackets. Is $s$ suppressed for brevity? Or have I misunderstood something?
- Section 3.3 Lines 415-420 Left Column: “At late times… smaller than $\sigma_D$.” I have no intuition for this result. Do the authors? If so, could they please add it?

---

> ### Author Rebuttal · Authors · 2025-04-01
>
> Thank you for the constructive feedback! We provide several additional results and clarifications here and will include them in the updated paper. Corresponding figures are in [the linked PDF](https://drive.google.com/file/d/1ZkobNTqh90nnUcunKqUT2Dyx3T4zAb5O/view).
>
> **Further evidence for the generality of supercollapse.**
> We focused on width scaling as it's the most well-studied dimension with extensive literature [1,2,3,4] on optimal learning rate and initialization scaling, which made finding the proper hyperparameter scaling needed for supercollapse much easier. Inspired by your question, we conducted a depth-scaling experiment by adding residual connections to our MLP (creating a well-defined depth scaling limit) and using depth-µP [5] to scale learning rate and initialization. Figure 1 (left) in the linked PDF shows depth scaling also produces supercollapse, confirming it's not width-specific; this plot is consistent with our theory linking supercollapse to power law compute-loss pareto frontiers.
>
> In addition, Figure 1 (middle, right) demonstrates supercollapse occurring in two additional data modalities: chess game string representations and chemical structures, further establishing its generality.
>
> **Realistic scaling recipes that don’t admit supercollapse.**
> We expect supercollapse to break down when hyperparameters such as the learning rate are not properly scaled with the model. In Figure 3 of the linked PDF, we ablate µP while keeping learning rate constant for all widths, which makes the learning rate increasingly suboptimal as width scales [4]. Without µP the rescaled curves don’t collapse.
>
>
> **Practical relevance of supercollapse.**
> Exploring supercollapse's practical applications is an exciting direction for future work! Our primary goal in this work is to establish supercollapse as a novel phenomenon across multiple data modalities and architectures and to show how studying loss curve universality enhances our understanding of scaling.
>
> That said, we believe this work already provides compelling evidence of supercollapse's practical value. Figures 3(a) and 4 in the paper show how absent supercollapse reveals errors in estimated data exponent $\gamma$. Similarly, MLP without µP (Figure 3, linked PDF) shows comparable performance to MLP with μP on the loss-compute plot except for the largest model, but the absence of supercollapse reveals scaling errors. Supercollapse is a powerful diagnostic tool, making errors more detectable than when evaluating final losses alone.
>
> **Methodology for fitting optimal training compute.**
> We estimate optimal compute by training models for a fixed number of steps (7e5 for transformers, 1e6 for MLPs) and identifying the compute-optimal point. We find models with lowest loss at each compute value on a logarithmic grid to create the compute-loss Pareto frontier (Figure 2). Our discrete model sizes mean each model is optimal for a range of compute values, producing multiple y-values per x-value in Figure 7 (Appendix C.1). We fit a power law to these points using log-log linear regression to estimate optimal compute per model size, similar to Chinchilla’s Figure 1 [6].
> We use constant schedule to efficiently sweep compute/token budgets, rather than running separate experiments for each budget as in Chinchilla. This reduces experimental costs, a strategy also used in [7] for LLM scaling law studies.
>
> **Additional clarifications**
>
> * $\theta$ refers to the fraction of optimal compute.
> * Figure 4 in linked PDF shows collapse deviation std dev across 5 trials. Colors indicate width (now in the legend).
> * $\Delta < \sigma$ for an $O(1)$ interval isn't trivial as $\Delta$ can grow quickly away from $\theta=1$, as we showed for constant schedule in Figure 6 of the paper.
> * SGD noise scales as $\Theta(\eta/B)$ (see Eq 33, Appendix E). Annealing LR thus reduces noise similar to increasing batch size.
> * Intuition for L415-420: variance of noise difference between timepoints is smaller than individual variances when sufficiently correlated.
> * We expanded $P_0$ range for sensitivity analysis in Figure 5 (linked PDF).
> * We suppressed $s$ in Eq 2 for brevity.
> * Outliers in Figure 9(a) occur when $\Delta\eta$ approaches 0 in multi-cycle cosine schedule, making $\hat{K}$ divergent and not well-defined.
>
> [1] Yang et al. "Feature learning in infinite-width neural networks."
>
> [2] Yang et al. "Tensor programs ivb: Adaptive optimization in the infinite-width limit."
>
> [3] Bordelon et al. "Self-consistent dynamical field theory of kernel evolution in wide neural networks."
>
> [4] Everett et al. "Scaling exponents across parameterizations and optimizers."
>
> [5] Yang et al. "Feature learning in infinite-depth neural networks."
>
> [6] Hoffmann et al. "Training compute-optimal large language models."
>
> [7] McLeish, Sean, et al. "Gemstones: A Model Suite for Multi-Faceted Scaling Laws."

---

### Decision · Program_Chairs · 2025-05-01

**Decision:**

Accept (oral)

**Comment:**

The paper introduces a novel phenomenon called supercollapse, where loss curves from compute-optimal neural network training collapse to a single curve (after an affine rescaling).

The paper provides a theoretical analysis (e.g., Theorem 3.1, analysis of power-law Pareto frontiers) to explain why supercollapse occurs, as well as experiments using transformers and MLPs across different learning rate schedules.

The reviewers raised some concerns over the limited variation in model widths and noted some minor issues but they generally regard the paper as solid and thorough, with most scoring it in the strong accept to accept range. I therefore recommend acceptance.